# SARS-CoV-2 ORF6 disrupts nucleocytoplasmic trafficking to advance viral replication

Yoichi Miyamoto [1✉], Yumi Itoh[2], Tatsuya Suzuki[2], Tomohisa Tanaka[3], Yusuke Sakai [4], Masaru Koido [5], Chiaki Hata[6], Cai-Xia Wang[6], Mayumi Otani[1], Kohji Moriishi [3,7], Taro Tachibana[6,8], Yoichiro Kamatani[9], Yoshihiro Yoneda[10], Toru Okamoto [2✉] & Masahiro Oka [1]

Severe acute respiratory syndrome coronavirus 2 (SARS-CoV-2) ORF6 is an antagonist of interferon (IFN)-mediated antiviral signaling, achieved through the prevention of STAT1 nuclear localization. However, the exact mechanism through which ORF6 prevents STAT1 nuclear trafficking remains unclear. Herein, we demonstrate that ORF6 directly binds to STAT1 with or without IFN stimulation, resulting in the nuclear exclusion of STAT1. ORF6 also recognizes importin α subtypes with different modes, in particular, high affinity to importin α1 but a low affinity to importin α5. Although ORF6 potentially disrupts the importin α/importin β1-mediated nuclear transport, thereby suppressing the nuclear translocation of the other classical nuclear localization signal-containing cargo proteins, the inhibitory effect of ORF6 is modest when compared with that of STAT1. The results indicate that the drastic nuclear exclusion of STAT1 is attributed to the specific binding with ORF6, which is a distinct strategy for the importin α1-mediated pathway. Combined with the results from a newly-produced replicon system and a hamster model, we conclude that SARS-CoV-2 ORF6 acts as a virulence factor via regulation of nucleocytoplasmic trafficking to accelerate viral replication, resulting in disease progression.

[1] Laboratory of Nuclear Transport Dynamics, National Institutes of Biomedical Innovation, Health and Nutrition (NIBIOHN), Osaka, Japan. [2] Institute for Advanced Co-Creation Studies, Research Institute for Microbial Diseases, Osaka University, Osaka, Japan. [3] Department of Microbiology, Graduate School of Medicine, University of Yamanashi, Yamanashi, Japan. [4] Department of Pathology, National Institute of Infectious Diseases, Tokyo, Japan. [5] Department of Cancer Biology, Institute of Medical Science, The University of Tokyo, Tokyo, Japan. [6] Cell Engineering Corporation, Osaka, Japan. [7] Division of Hepatitis Virology, Institute for Genetic Medicine, Hokkaido University, Hokkaido, Japan. [8] Department of Bioengineering, Graduate School of Engineering, Osaka Metropolitan University, Osaka, Japan. [9] Department of Computational Biology and Medical Sciences, Graduate School of Frontier Sciences, The University of Tokyo, Tokyo, Japan. [10] National Institutes of Biomedical Innovation, Health and Nutrition (NIBIOHN), Osaka, Japan. ✉email: ymiyamoto@nibiohn.go.jp; toru@biken.osaka-u.ac.jp

The coronavirus disease 2019 (COVID-19) pandemic is caused by severe acute respiratory syndrome coronavirus 2 (SARS-CoV-2), which is a single-strand RNA virus belonging to the *Coronaviridae* family[1–3]. The genome of SARS-CoV-2 is ~29.7 kb long with short untranslated regions (UTR) at the 5′ and 3′ termini, and encodes nonstructural (nsp1–16), structural (spike [S], envelope [E], membrane [M], and nucleocapsid [N]), and accessory proteins (ORF3a, ORF3b, ORF6, ORF7a, ORF7b, ORF8, and ORF10)[4,5].

Among them, ORF6 is a small protein of approximately 7 kDa, which consists of 61 amino acids and exhibits a 69% homology with the SARS-CoV ORF6, from which it differs due to a two amino acid deletion at the C-terminus[6]. Several studies have recently shown that both the SARS-CoV and SARS-CoV-2 ORF6 proteins antagonize the host innate immune system via the Janus activated kinase 1 (JAK1)- and JAK2-signal transducers, and activators of transcription (STAT)[6–12]. STAT1 is a key mediator of cytokine-induced gene expression as it is activated by cytokines including type I and type II interferons (IFNs)[13,14]. Activation of JAKs associated with type I IFN receptor results in the tyrosine phosphorylation of STAT1 (PY-STAT1), leading to the formation of a STAT1-STAT2 heterodimer, while Type I interferon (IFN-α or -β) and type II interferon (IFN-γ) induce the formation of the PY-STAT1 homodimers. Both the hetero- and homo-dimer STAT1 complexes translocate to the nucleus to bind the IFN-stimulated response elements (ISRE) or IFN-γ-activated site (GAS)[13,14]. Previous studies have reported that ORF6 inhibits the nuclear transport of PY-STAT1 to suppress primary interferon signaling[7,10,11].

Constitutive and signal-dependent protein transport through nuclear pore complexes (NPCs) embedded in the nuclear envelope are mediated by members of the importin (also known as karyopherin) superfamily[15–18]. The process of protein import into the nucleus commonly involves the recognition of a classical nuclear localization signal (cNLS) by the importin α/importin β1 heterodimer[19,20]. Cargos containing the cNLS are recognized by the adaptor molecule importin α (also known as karyopherin α: KPNA). Following the entrance of the cNLS-containing cargo/importin α/β1 trimeric complex into the nucleus through the NPCs, the cargo is released from importin α by binding of GTP-bound small GTPase Ran (RanGTP) to importin β1[19–21]. After the complex dissociation, importin α is exported to the cytoplasm by cellular apoptosis susceptibility gene product (CAS, also known as CSE1L) with RanGTP, and the importin β1/RanGTP complex also returns to the cytoplasm, where it is reused in the subsequent rounds of transport[15,21].

Seven importin α proteins have been identified in humans, whereas six have been discovered in mice[18,21,22]. Based on sequence similarity, each importin α protein is assigned to one of three conserved subfamilies. In humans, clade 1 consists of importin α5 (encoded by the *KPNA1* gene), importin α6 (*KPNA5*), and importin α7 (*KPNA6*). Clade 2 consists of importin α1 (*KPNA2*) and importin α8 (*KPNA7*), and clade 3 consists of importin α3 (*KPNA4*) and importin α4 (*KPNA3*)[21]. The members of each subfamily display cargo specificity and are differentially expressed in different tissues and cell types[21–24]. Differences in the usage of "importin α" or "KPNA" and the number of proteins in humans have often been a source of confusion. Therefore, in the present study, we uniformly use the human nomenclature for "importin α" to refer to the protein and use the italic term "*KPNA*" to refer to the gene, while the normal term "KPNA" is described according to the way in which it is used in the citation.

It has already been demonstrated that nuclear transport of the PY-STAT1 as a homodimer or a heterodimer with STAT2 is mediated by specific clade 1 subtypes of importin α such as importin α5 (referred to as KPNA1 in some papers)[25–28]. According to the several studies on the viral proteins that inhibit nuclear transport of PY-STAT1, SARS-CoV ORF6 has been reported to tether KPNA2, but not KPNA1, to ER to sequester importin β1 into endoplasmic reticulum (ER)/Golgi apparatus, resulting in the suppression of nuclear import of PY-STAT1[7]. Recently, SARS-CoV-2 ORF6 has also been shown to interact with KPNA2 and KPNA1[10,11], further supporting its interference with the nuclear transport of PY-STAT1. In addition, SARS-CoV-2 ORF6 has been reported to interact with the NPC components Nup98 and RAE1 to inhibit the nuclear transport of a broad range of proteins, including PY-STAT1[10,29,30]. However, the exact molecular mechanisms of the effects of SARS-CoV-2 ORF6 on the nucleocytoplasmic trafficking remains largely unclear. In particular, our understanding of that ORF6 induces the remarkable nuclear exclusion of STAT1 with high specificity remains poor.

In the present study, we characterized the effect of SARS-CoV-2 ORF6 on nucleocytoplasmic protein transport. We observed that ORF6 directly binds to STAT1 to suppress IFN-induced nuclear localization and nuclear shuttling in the absence of IFN-stimulation. In addition, the direct binding of ORF6 to importin α1 significantly reduces cNLS-cargo transport, whereas the inhibitory effect of ORF6 is more prominent for STAT1. The results reveal that ORF6 has distinct strategies of disrupting nucleocytoplasmic trafficking of STAT1 and importin α1-mediated cargos. Finally, the results from a newly produced replicon system and a hamster model demonstrate that ORF6 acts as a virulence factor via the regulation of nucleocytoplasmic trafficking to accelerate viral replication.

## Results

**ORF6 inhibits nuclear localization of STAT1 following IFN stimulation.** Several studies have already demonstrated that ORF6 inhibits nuclear localization of STAT1 in response to type-I IFN (IFN-α or -β) stimulation[7–11]. Here, we attempted to verify the inhibitory effects of SARS-CoV-2 ORF6 on a type-II IFN (IFN-γ)-activated STAT1. An AcGFP-fused ORF6 was transfected into HeLa cells, and the subcellular localization of PY-STAT1 was observed when the cells were stimulated with either IFN-β or IFN-γ. Whereas the PY-STAT1 was localized in the nucleus of AcGFP-transfected cells in response to each treatment, the distribution shifted markedly to the cytoplasm in AcGFP-ORF6-transfected cells (Fig. 1a, b). The fluorescence intensity ratio of the nucleus against the whole cells further supported the significant nuclear exclusion of the PY-STAT1 in the AcGFP-ORF6-transfected cells when stimulated by IFN-γ (Fig. 1c).

Subsequently, to confirm that the PY-STAT1 is excluded from the nucleus by ORF6, we evaluated the expression of PY-STAT1 downstream genes using quantitative RT-PCR (qRT-PCR). In comparison to the AcGFP-transfected control cells, the AcGFP-ORF6-transfected cells showed significant down-regulation of IFN-γ-inducible protein 10 (*IP-10*)[31] mRNA 6 h post-transfection or later, upon stimulation (Fig. 1d). To clarify whether the nuclear exclusion of PY-STAT1 by ORF6 affects the expression of the interferon gamma-activated sites (GAS) or the IFN-stimulated response element (ISRE)-containing gene, a luciferase assay was performed in Huh7 cells. The cells were transfected with a luciferase reporter plasmid including GAS or ISRE together with AcGFP or AcGFP-ORF6. We observed significant repression of the relative luciferase values in the AcGFP-ORF6-transfected cells compared to that in AcGFP-transfected cells (Fig. 1e (GAS) and f (ISRE); raw value graphs are presented in Supplementary Fig. 1a (GAS) and b (ISRE)). Finally, we confirmed that the relative luciferase value was significantly

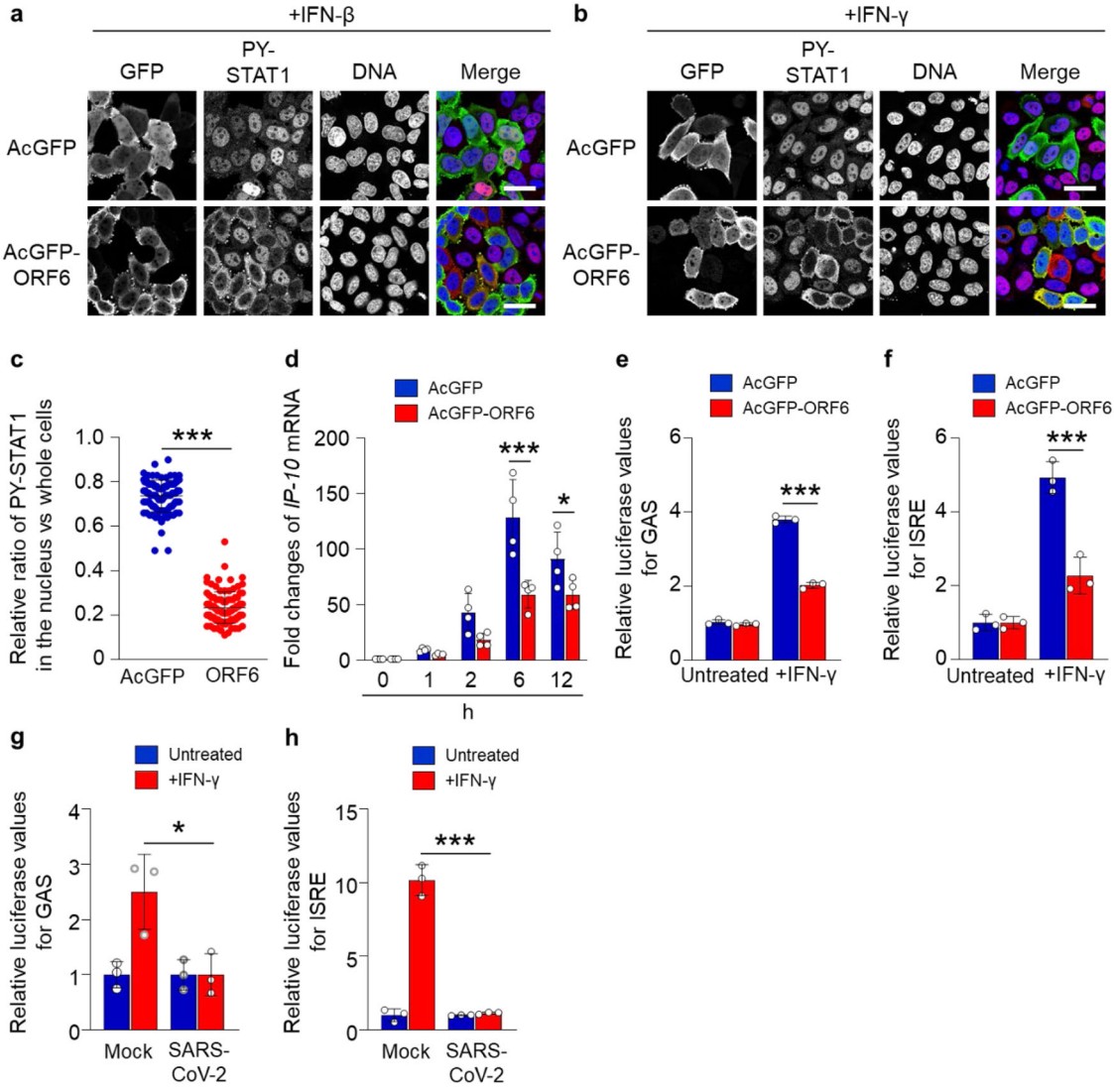

**Fig. 1 Inhibition of nuclear localization of PY-STAT1 by ORF6. a, b** Immunofluorescence of phosphorylated STAT1 (PY-STAT1) in HeLa cells transfected with AcGFP or AcGFP-ORF6 following interferon-β (IFN-β) **a** or IFN-γ **b** stimulation. GFP (green) and PY-STAT1 (red) were identified using specific antibodies. DAPI staining (blue) was used for DNA staining. Scale bars: 30 μm. **c** The graph represents the relative fluorescence values of PY-STAT1 in the nucleus compared to those of the whole cells (shown in **b**). Signal intensities of total 100 different nuclei from two independent experiments. ***$P < 0.001$, two-tailed Student's t-test. error bars represent SD. **d** qRT-PCR analysis of *IP-10* mRNA in AcGFP or AcGFP-ORF6-transfected HeLa cells at the described time points ($n = 4$ each). *$P < 0.05$, ***$P < 0.001$, two-way ANOVA. error bars represent SD. **e, f** Relative luciferase values of GAS-Luc **e** or ISRE-Luc **f** in AcGFP- or AcGFP-ORF6-transfected Huh7 cells following IFN-γ stimulation ($n = 3$ each). ***$P < 0.001$, one-way ANOVA. error bars represent SD. **g, h** Relative luciferase values of GAS-Luc **g** or ISRE-Luc **h** in VeroE6/TMPRSS2 cells infected with SARS-CoV-2 (NIID strain) following IFN-γ stimulation ($n = 3$ each). *$P < 0.05$, ***$P < 0.001$, one-way ANOVA. error bars represent SD.

suppressed when the SARS-CoV-2 infected cells were stimulated by IFN-γ (Fig. 1g (GAS) and h (ISRE)). The results indicate that SARS-CoV-2 ORF6 suppresses nuclear translocation of PY-STAT1 to inhibit the activation of STAT1-downstream genes.

**The C-terminal region of ORF6 aids the acceleration of viral replication.** Previously, the C-terminal region of ORF6 has been shown to involve in the inhibition of IFN response[8]. Hence, in line with previous findings, we validated the effects of ORF6 C-terminal mutations on the nuclear translocation of PY-STAT1. We observed that the ORF6 mutant with alanine replacing amino acids (a.a.) 49 to 52 (referred to as ORF6-M1) retained the inhibitory effect over PY-STAT1, while the other mutants with alanine replacing a.a. 53 to 55 (ORF6-M2) and 56 to 61 (ORF6-M3) did not retain their inhibitory functions (Fig. 2a–c).

To further assess the function of ORF6 in viral propagation, we established a replicon system in which we could evaluate the viral replication process by detecting Renilla luciferase (RLuc) (Fig. 2d). The replicon plasmid was transfected together with AcGFP or AcGFP-ORF6 in Huh7 cells and the RLuc values were analyzed 24 h post-transfection. We observed that the expression of WT ORF6 and ORF6-M1, but not ORF6-M2 and ORF6-M3, significantly enhanced viral replication (Fig. 2e), consistent with the effects on STAT1 nuclear localization. In addition, we found that IFN-β-mediated down-regulation of the viral replication was rescued by the presence of ORF6 (Supplementary Fig. 2), supporting the hypothesis that ORF6 antagonizes the IFN-signaling to enhance the viral replication through inhibition of the STAT1 nuclear trafficking. Hence, we next attempted to examine the relationships between ORF6 and STAT1. STAT1 expression was knocked down by siRNA transfection (Fig. 2f),

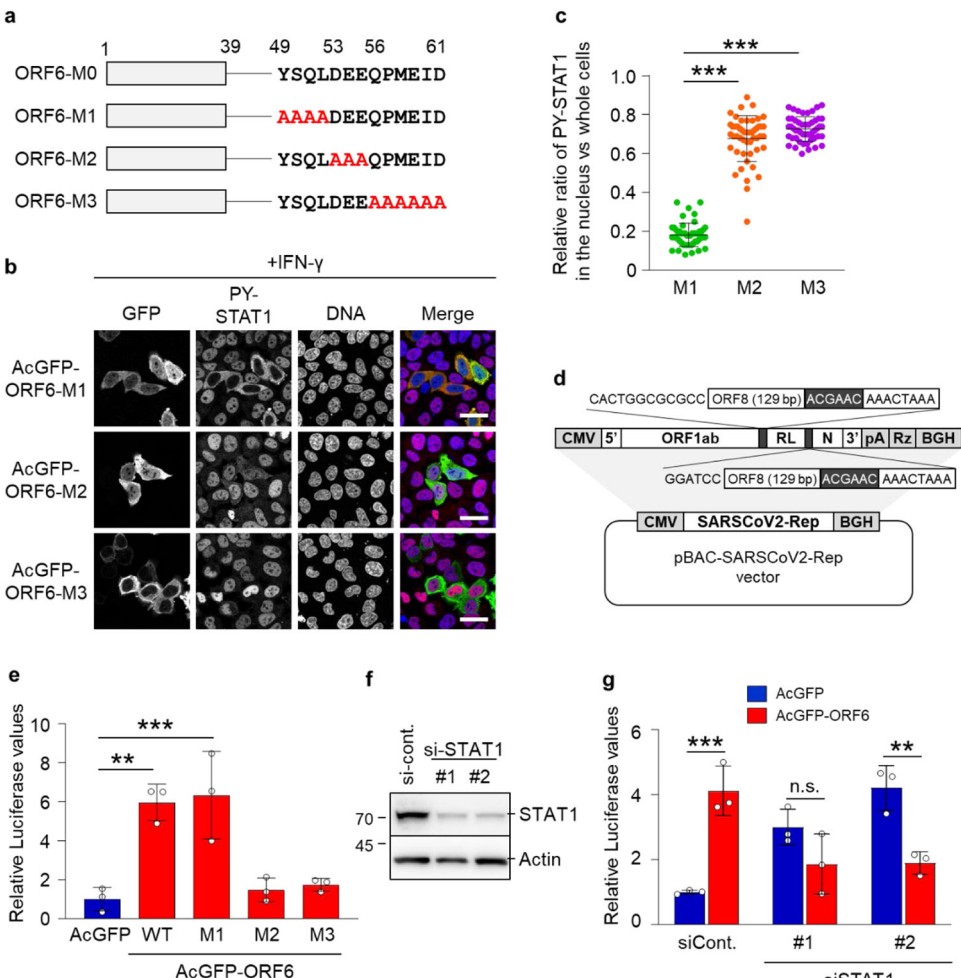

**Fig. 2 The C-terminal region of ORF6 participates in viral RNA replication. a** Schematic representation of ORF6 and its alanine substitution mutations within the C-terminus. ORF6-M0: ORF6 wild type; ORF6-M1: ORF6 with amino acids 49-52 substituted for alanine; ORF6-M2: ORF6 with amino acids 53-55 substituted for alanine; ORF6-M3: ORF6 with amino acids 56–61 substituted for alanine. **b** Immunofluorescence of PY-STAT1 in HeLa cells transfected with the AcGFP-ORF6 mutants following IFN- γ stimulation. GFP (green) and PY-STAT1 (red) were identified using specific antibodies. DAPI was used to stain the DNA (blue). Scale bars: 30 μm. **c** The graph represents the relative fluorescence values of PY-STAT1 in the nucleus compared to those of the whole cells in **b**. Signal intensities of total 50 nuclei from two independent experiments were measured. ***$P < 0.001$, one-way ANOVA. error bars represent SD. **d** Schematic representation of SARS-CoV-2 replicon DNA, pBAC-SCoV2-Rep. The genetic structure of the SARS-CoV-2 replicon is shown at the top of the panel. The dark shaded box indicates the core sequence of transcription regulating sequence. *CMV* cytomegalovirus promoter, *RL* Renilla luciferase gene, *pA* a synthetic poly(A) tail, *Rz* hepatitis delta virus ribozyme, *BGH* bovine growth hormone polyadenylation sequence. **e** Relative luciferase values for ORF6 wild type (WT) or each mutant in replicon ($n = 3$). **$P < 0.01$, ***$P < 0.001$, one-way ANOVA. error bars represent SD. **f** Huh7 cells were transfected with siRNAs for STAT1 (si-STAT1) or control siRNA (si-cont.), and cell lysates were subjected to western blotting. Two different siRNAs (shown as #1 and #2) for STAT1 were used for this knockdown experiment. The proteins were detected using specific antibodies for STAT1 or Actin. Values are kDa. **g** Huh7 cells were transfected with the indicated siRNAs overnight and then transfected with pBAC-SCoV2-Rep together with AcGFP or AcGFP-ORF6 for 24 h. Relative luciferase values were determined by luciferase assay ($n = 3$). **$P < 0.01$, ***$P < 0.001$, n.s.: not significant, two-way ANOVA. error bars represent SD.

and then the cells were transfected with the replicon plasmid together with AcGFP or AcGFP-ORF6, respectively. As a result, the enhancement of viral replication in the cells expressing AcGFP-ORF6 was impaired by the knockdown of STAT1 expression (Fig. 2g). The results suggest that the C-terminal regions (a.a. 53 to 61) of ORF6 inhibit interferon signaling by disrupting nuclear localization of PY-STAT1, resulting in the enhancement of SARS-CoV-2 viral replication.

**ORF6 changes the subcellular localization of STAT1 in an IFN-independent manner**. Next, to characterize the interplay between ORF6 and STAT1 in more detail, we analyzed the subcellular distribution of Flag-tagged STAT1 in AcGFP-ORF6-transfected HeLa cells. In addition to the WT ORF6, the in-frame 9 a.a.

deletion mutant (loss of a.a. 22 to 30; referred to as ORF6Δ9), which has been reported in previous studies[32], was also examined. Under non-stimulated conditions (untreated), Flag-STAT1 was mainly localized in the cytoplasm in either AcGFP-, AcGFP-ORF6-, or AcGFP-ORF6Δ9-transfected cells (Fig. 3a, c). Upon IFN-γ stimulation, Flag-STAT1 was distributed into the nucleus in the AcGFP-transfected cells (Fig. 3b, d). In contrast, it was retained in the cytoplasm in AcGFP-ORF6 WT- and Δ9-transfected cells. Notably, statistical analysis revealed that even under non-stimulated conditions, the cytoplasmic intensities of Flag-STAT1 were significantly higher in the AcGFP-ORF6 WT- or Δ9-transfected cells, when compared with those observed in the AcGFP-transfected control cells (Fig. 3c), suggesting that STAT1 may shuttle between the nucleus and the cytoplasm in the

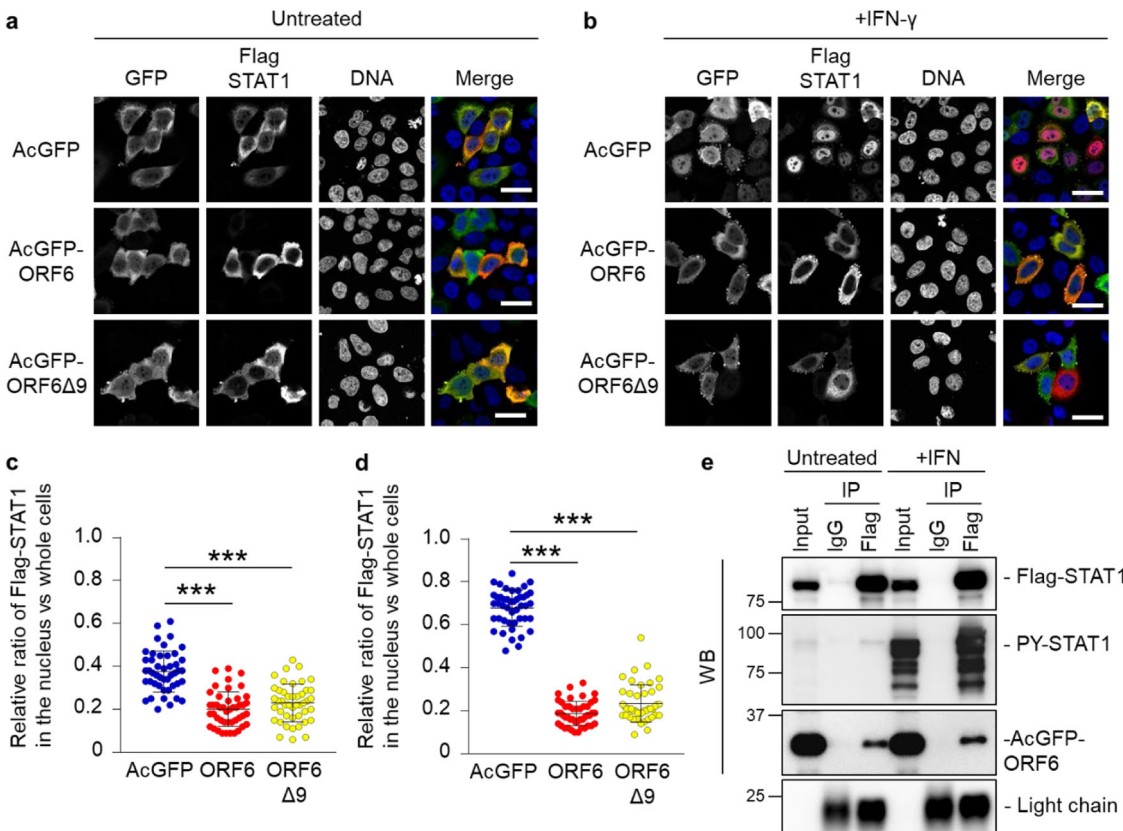

**Fig. 3 ORF6 changes the subcellular localization of STAT1 in an IFN-independent manner. a, b** Immunofluorescence of Flag-STAT1 in HeLa cells transfected with AcGFP, AcGFP-ORF6 WT, or AcGFP-ORF6Δ9 under an untreated condition **a** or following IFN-γ stimulation **b**. Anti-GFP and anti-Flag antibodies were used for the detection of AcGFP (green) and Flag-STAT1 (red), respectively. DAPI was used to stain the DNA (blue). Scale bars: 30 μm. **c, d** The graphs represent the relative fluorescence values of Flag-STAT1 in the nucleus compared to those of the whole cells under an untreated condition in **a** or following IFN-γ stimulation in **b**, respectively. Signal intensities from total 45 nuclei from two independent experiments. ***$P < 0.001$, one-way ANOVA. error bars represent SD. **e** Immunoprecipitation (IP) of Flag-STAT1 from HEK293 cells transfected with AcGFP-ORF6 which is tagged the HA sequence following stimulation with IFN-γ (+IFN). Normal mouse IgG was used as a negative control for IP. Input was 1/130 dilution of cell lysates used for the reaction. Flag-STAT1 and AcGFP-ORF6 were detected by either anti-Flag or anti-HA antibodies, respectively (WB). The stimulation of IFN-γ was identified by the detection of PY-STAT1 using anti-PY-STAT1 antibody. Light chain indicates the precipitated antibodies.

absence of IFN stimulation and be trapped in the cytoplasm by ORF6. Therefore, to address the potential interaction between ORF6 and non-activated STAT1, we first performed an immunoprecipitation (IP) assay using AcGFP-ORF6- and Flag-STAT1-transfected cells. As a result, AcGFP-ORF6, which was detected by anti-HA antibody, was precipitated with Flag-STAT1 even in the absence of IFN stimulation (Fig. 3e).

**ORF6 directly binds to STAT1 through the C-terminus.** Subsequently, to determine whether ORF6 directly binds to STAT1, bacterially purified recombinant STAT1 protein was incubated with either recombinant GST-GFP or GST-GFP-fused ORF6 full-length protein (GST-GFP-ORF6). Figure 4a shows that STAT1 was directly bound to GST-GFP-ORF6. To address whether the binding of ORF6 to STAT1 is mediated by the amino acids 49-61 in the C-terminal region (referred to as M0 in the present study, see Fig. 2a), we produced a recombinant protein consisting of the 13 a.a. (M0) which was fused to GST and GFP (GST-M0-GFP). As shown in Fig. 4b, the pull-down assay clearly revealed that STAT1 bound to the GST-M0-GFP protein. The result is reinforced by the additional evidence that the C-terminus deletion mutant of ORF6 showed dramatically decreased binding to STAT1 (Fig. 4c). We also confirmed that GST-GFP-ORF6 clearly bound to endogenous PY-STAT1 in HeLa cells stimulated by IFN-γ (Fig. 4d). Additional GST-pull-down assays demonstrated

that the interaction with PY-STAT1 occurred through the C-terminal sequence of ORF6, and this binding was abolished with the M2 and M3 mutants (Fig. 4e). Furthermore, to address whether only the C-terminus sequence could inhibit the nuclear localization of STAT1, the GST-GFP fusion mutants were microinjected into the cytoplasm of Huh7 cells, and then, the subcellular distribution of PY-STAT1 was observed. As shown in Fig. 4f, g, the M0, M1, and M2 proteins showed localization not only to the cytoplasm but also to the nucleus, whereas the GST-GFP control protein was localized only to the cytoplasm. In contrast, the M3 protein was observed in the cytoplasm. With IFN stimulation, the nuclear migration of PY-STAT1 was suppressed in the M0- and M1-mutant-injected cells (Fig. 4g), consistent with that observed for the AcGFP-ORF6 M0 and M1 mutants (Fig. 2b), whereas it could not completely achieve this nuclear exclusion, unlike the full-length ORF6 protein was expressed in cells. Overall, we concluded that ORF6 directly binds to STAT1 through the C-terminus, disrupting the nuclear trafficking of STAT1.

**ORF6 alters subcellular distribution of importin α subtypes distinctly.** Previous studies have shown that SARS-CoV- and SARS-CoV-2-associated inhibition of nuclear translocation of STAT1 is accomplished through interaction between ORF6 and importin α1 (referred to as KPNA2 in the original paper)[7,11].

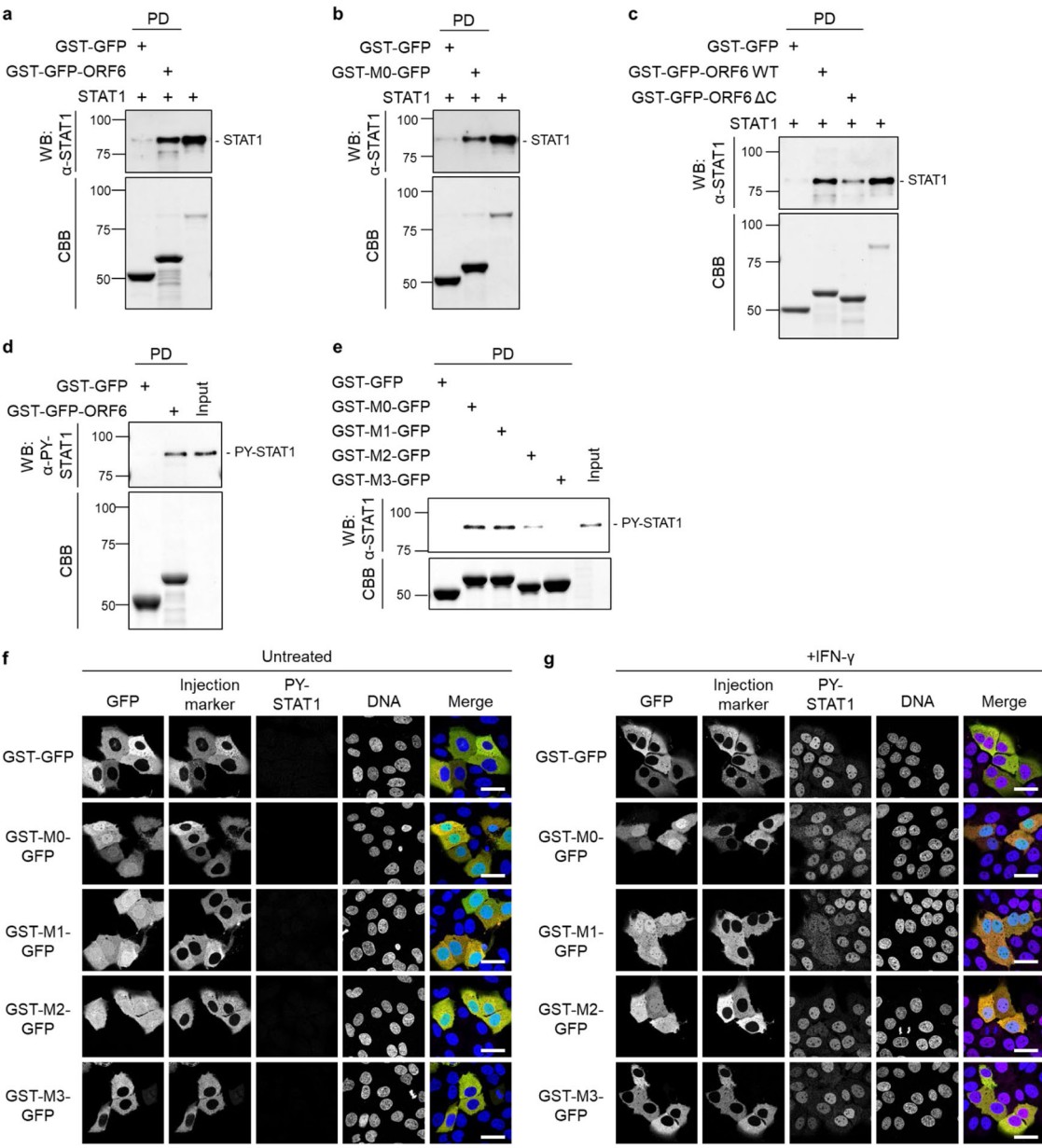

**Fig. 4 The C-terminal sequence of ORF6 disrupts the nuclear localization of STAT1. a** The bacterially purified STAT1 recombinant protein was incubated with either GST-GFP or GST-GFP-ORF6 immobilized on glutathione Sepharose beads (GST-beads) for 1 h, and the pulled-down protein was collected (indicated as PD). STAT1 was detected with anti-STAT1 antibody (WB). The bottom panel represents the proteins bound to the GST-beads and stained with Coomassie Brilliant Blue (CBB). The right lane was loaded with the STAT1 protein (1/15,000 dilution of the reaction) as an input protein. Values are kDa. **b** The STAT1 recombinant protein was incubated with either GST-GFP or GST-GFP fused with the C-terminal peptide of ORF6 wild type (49-61 amino acids; GST-M0-GFP) immobilized on GST-beads for 1 h, and then a pulled-down protein was collected (PD). STAT1 was detected using anti-STAT1 antibody (WB). The bottom panel represents the proteins bound to the beads and stained with CBB. The right lane loaded with the STAT1 protein (1/10,000 dilution of the reaction) as an input protein. Values are kDa. **c** The STAT1 recombinant protein was incubated with either GST-GFP, GST-GFP-ORF6 wild type (WT), ore GST-GFP-ORF6 C-terminal deletion mutant lacking a.a. 49–61 (ΔC) immobilized on GST-beads for 1 h, and then a pulled-down protein was collected (PD). STAT1 was detected using anti-STAT1 antibody (WB). The bottom panel represents the proteins bound to the beads and stained with CBB. The right lane loaded with the STAT1 protein (1/15,000 dilution of the reaction) as an input protein. Values are kDa. **d** Lysates from HeLa cells stimulated with IFN-γ were incubated with either GST-GFP or GST-GFP-ORF6 immobilized on GST-beads for 1 h, and the pulled-down proteins were collected (PD). PY-STAT1 was detected with a specific antibody (WB). The bottom panel represents the proteins bound to the beads and stained with CBB. The input was a 1/500 dilution of cell lysates used for the reaction. Values are kDa. **e** Lysates from HeLa cells stimulated with IFN-γ were incubated with GST-GFP, GST-M0-GFP, GST-M1-GFP, GST-M2-GFP, or GST-M3-GFP, which were immobilized on GST-beads for 1 h, and the pulled-down proteins were collected (PD). PY-STAT1 was detected with a specific antibody (WB). The bottom panel represents the proteins bound to the beads and stained with CBB. The input was a 1/1,000 dilution of cell lysates used for the reaction. **f**, **g** Huh7 cells were microinjected with either GST-GFP, GST-M0-GFP, GST-M1-GFP, GST-M2-GFP, or GST-M3-GFP proteins (green), as well as a Fluorescence-conjugated antibody (red) as an injection marker, into the cytoplasm. The injected cells were incubated **f** or treated without IFN-γ **g** for 30 min, and PY-STAT1 (magenta) was detected using specific antibodies. DAPI was used for DNA staining (blue). Scale bars: 30 μm.

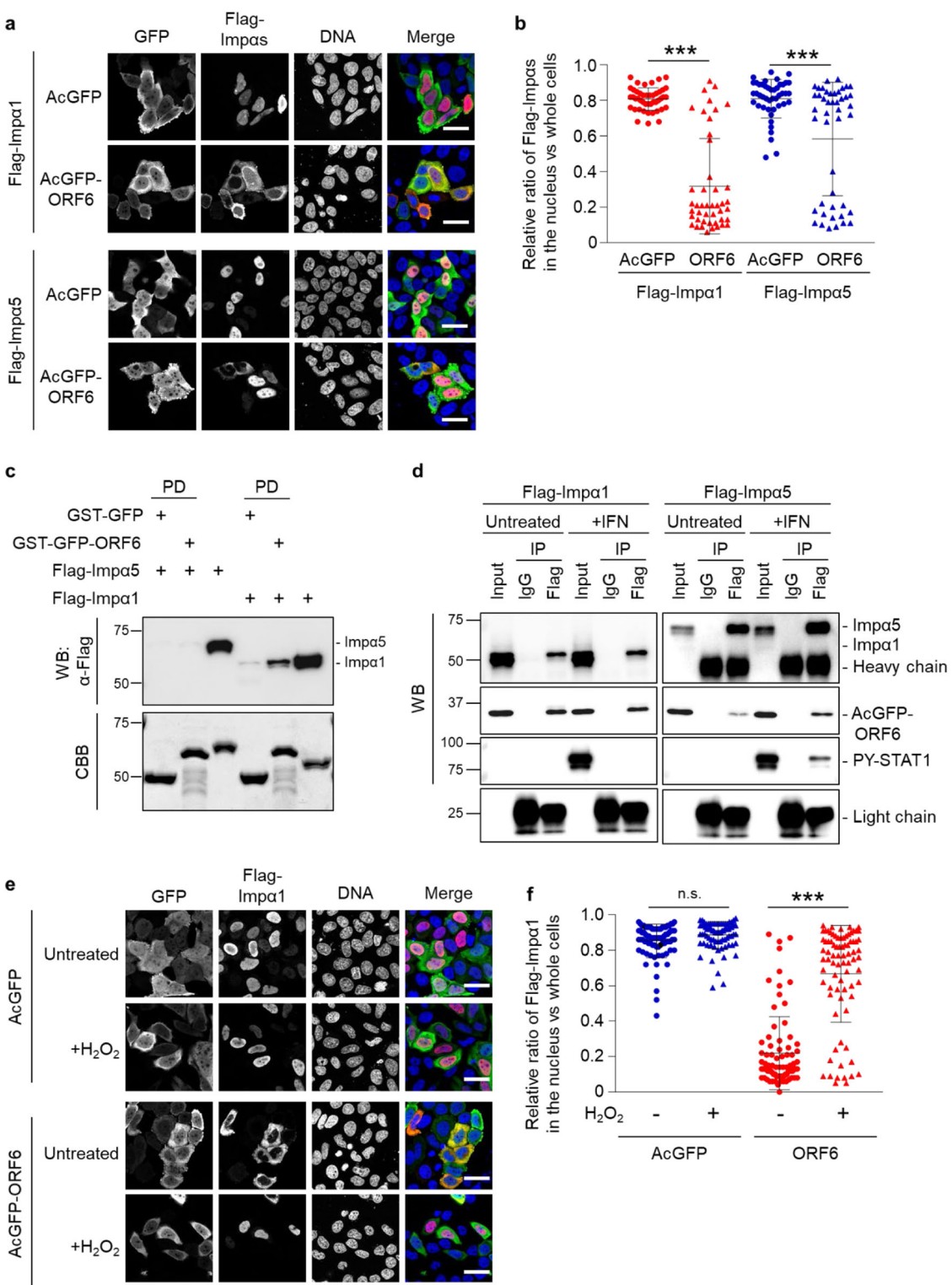

However, as described above, ORF6 binds directly to STAT1 in the absence of IFN stimulation in the cytoplasm, which raises a possibility that the binding of ORF6 to importin α proteins might not be required for the inhibition of nuclear accumulation of STAT1. Therefore, we explored the interplay between ORF6 and importin α proteins. Since ORF6 has been shown to alter the distribution of importin α1 (KPNA2) and/or importin α5 (KPNA1)[7,10], we first validated the findings concerning all human importin α subtypes. Consistent with the previous reports, in AcGFP-transfected cells (control), the overexpressed Flag-

importin α proteins were mainly localized in the nucleus (Fig. 5a, b and Supplementary Fig. 3). In contrast, in AcGFP-ORF6-transfected cells, the localization of Flag-importin α1, α3, α4, α6, and α8 remarkably shifted to the cytoplasm, while Flag-importin α5 and α7 were still mostly localized in the nucleus (Fig. 5a, b and Supplementary Fig. 3). The analysis of nuclear fluorescence intensity ratios clearly shows that in AcGFP-ORF6-transfected cells, the nuclear distribution of Flag-importin α1 was drastically shifted to the cytoplasm, while the number of cells in which Flag-importin α5 is localized in the cytoplasm was limited

**Fig. 5 ORF6 affects the subcellular localization of importin α proteins. a** Immunofluorescence of Flag-importin α1 (Flag-Impα1) and Flag-importin α5 (Flag-Impα5) in HeLa cells transfected with AcGFP or AcGFP-ORF6. Anti-GFP and anti-Flag antibodies were used for detection of AcGFP (green) and Flag-importin αs (red), respectively. DAPI was used to stain the DNA (blue). Scale bars: 30 μm. **b** The graph represents the relative fluorescence values of Flag-importin αs in the nucleus compared to those of the whole cells in **a**. Signal intensities of total 50 nuclei from two independent experiments were measured. ***$P < 0.001$, two-tailed Student's t-test. error bars represent SD. **c** The Flag-Impα1 or Flag-Impα5 recombinant proteins were incubated with GST-GFP and GST-GFP-ORF6 immobilized on GST-beads for 1 h, and then a pulled-down protein was collected (PD). The importin α proteins were detected using an anti-Flag antibody (WB). The bottom panel represents the proteins bound to the beads and stained with CBB. Either the Flag-Impα1 protein or the Flag-Impα5 protein was loaded as an input protein (1/20,000 dilution of the reaction). Values are kDa. **d** Immunoprecipitation (IP) of Flag-Impα1 or Flag-Impα5 from AcGFP-ORF6-transfected HEK293 cells following IFN-γ stimulation. HEK293 cells transfected with AcGFP-ORF6 and each Flag-importin α were treated with IFN-γ (+IFN) for 1 h. The cell lysates were incubated with anti-Flag antibody for IP. Each importin α was detected by the specific antibodies (WB). Anti-HA and anti-PY-STAT1 antibodies were used for detection of AcGFP-ORF6 and endogenous PY-STAT1, respectively. Normal mouse IgG was used as a negative control for IP. Light chain indicates the antibodies precipitated. Input was 1/400 dilution of cell lysates used for the reaction. Heavy chain appeared in the Flag-Impα5 IP sample. **e** Immunofluorescence of Flag-Impα1 in HeLa cells transfected with AcGFP or AcGFP-ORF6 following hydrogen peroxide (200 μM $H_2O_2$) treatment. Anti-GFP and anti-Flag antibodies were used for the detection of AcGFP (green) and Flag-Impα1 (red), respectively. DAPI was used to stain the DNA (blue). Scale bars: 30 μm. **f** The graph represents the relative fluorescence values of Flag-Impα1 in the nucleus compared to those of the whole cells in **e**. Signal intensities of total 80 nuclei from two independent experiments were measured. ***$P < 0.001$, n.s.: not significant, two-tailed Student's t-test. error bars represent SD.

compared with that of Flag-importin α1 (Fig. 5b). The data indicate that ORF6 has distinct effects on each importin α subtype.

**Importin α1 shuttles between the nucleus and the cytoplasm in the presence of ORF6.** Next, we examined whether ORF6 directly binds to importin α proteins. Purified recombinant GST-GFP-ORF6 was incubated with Flag-importin α1 or Flag-importin α5, respectively, and then the GST-proteins were pulled down using glutathione Sepharose beads (GST-beads) to determine whether the Flag-importin α proteins were co-precipitated or not. The pull-down assay results indicated that ORF6 is certainly bound to Flag-importin α1, while only a faint band was detected for Flag-importin α5 (Fig. 5c). We also clarified the binding of ORF6 and importin αs using an immunoprecipitation assay. HEK293 cells were transfected with AcGFP-ORF6 and either Flag-importin α1 or Flag-importin α5, following IFN-γ stimulation, and then the Flag-importin α protein was precipitated by anti-Flag antibody. As shown in Fig. 5d, AcGFP-ORF6 apparently showed the binding with Flag-importin α1 with or without the interferon stimulation, while PY-STAT1 could not be identified even in the stimulated condition. Conversely, the binding status for importin α5 depended on the IFN stimulation. In the unstimulated condition (Untreated), AcGFP-ORF6 was certainly but faintly detected in the sample precipitated by Flag-importin α5. However, the band intensity of AcGFP-ORF6 was clearly increased in response to the IFN stimulation and PY-STAT was also detected (Fig. 5d), which coincided with the previous report that PY-STAT1 can be recognized with importin α5 (referred to as NPI-1 in the original paper), not importin α1 (referred to as Rch1)[25]. Overall, there are importin α subtype specificities, with ORF6 mainly binding to importin α1, in contrast to the low affinity for importin α5, corresponding to the subcellular distribution change for each importin α.

Direct binding of ORF6 to importin α1 suggests that ORF6 may inhibit the mobility of importin α1 to tether it in the cytoplasm, as reported previously with SARS-CoV ORF6[7]. Indeed, the cytoplasmic localization of endogenous importin α1 was accelerated in ORF6-transfected cells (Supplementary Fig. 4a, b), and was not affected by stimulation with IFNs (Supplementary Fig. 4c–e). To examine the mobility of importin α1 in ORF6-expressed cells, we focused on importin α1 accumulation in the nucleus in response to cellular stresses such as oxidative stress[33,34]. It has been clearly demonstrated that while in unstressed cells importin α1 shuttles between the nucleus and the cytoplasm, in stressed cells, importin α1 accumulates in the

nucleus due to the inhibition of RanGTP-dependent nuclear export of importin α by a collapse in the RanGTP gradient[33,34]. Therefore, we speculated that if importin α1 is tethered in the cytoplasm by ORF6, its nuclear accumulation cannot be observed under stress conditions. Hence, HeLa cells were transfected with AcGFP-ORF6 and Flag-importin α1, and then treated with 200 μM of Hydrogen peroxide ($H_2O_2$) for 1 h. Under oxidative stress conditions, the cytoplasmic localization of Flag-importin α1 was markedly shifted to the nucleus in the ORF6-transfected cells (Fig. 5e, f). Moreover, clear nuclear localization of endogenous importin α1 was observed under oxidative stress, even in the presence of ORF6 (Supplementary Fig. 5a, b). Conversely, no alteration in the distribution of Flag-STAT1 was observed (Supplementary Fig. 5c, d). Therefore, the results indicate that although importin α1 seems to be tethered in the cytoplasm of the ORF6-transfected cells, it retains its original capacity to move between the nucleus and the cytoplasm even in the ORF6-transfected cells. This model is supported by the results that the subcellular localization of importin β1 did not dramatically alter in the ORF6-transfected cells (Supplementary Fig. 6a, b), while CAS, the export different for importin α, apparently shifted to the nucleus (Supplementary Fig. 6c, d). Taken together, it is most likely that the nuclear exclusion of STAT1 by SARS-CoV-2 ORF6 is not due to the cytoplasmic tethering of importin α1, which could be differed from that of SARS-CoV ORF6[7].

**ORF6 negatively regulates the importin α/importin β1 pathway.** We subsequently attempted to determine whether ORF6 directly affects the importin α1-mediated nuclear transport pathway or not. For this, HeLa cells were transfected with mCherry-fused SV40T-NLS (mCherry-NLS) together with AcGFP or AcGFP-ORF6, and then the nuclear intensities of the fluorescent substrate were measured. Although the majority of mCherry-NLS was observed in the nucleus of AcGFP-ORF6-transfected cells, its cytoplasmic proportion was significantly increased compared to that observed in the AcGFP-transfected control cells (Fig. 6a, b).

Next, we compared the binding ability of importin α1 to the cNLS-containing cargo (GST-NLS-GFP) and ORF6 (GST-GFP-ORF6). The GST pull-down assay showed that importin α1 more efficiently interacted with cNLS than ORF6 (Fig. 6c). To assess the inhibitory effects of ORF6 on the importin α/β1-mediated nuclear transport of the cNLS-cargo, a digitonin-permeabilized semi-intact nuclear transport assay was performed. We observed that the addition of excess amounts of ORF6 significantly inhibited the nuclear translocation of GST-NLS-mRFP (Fig. 6d, e), suggesting

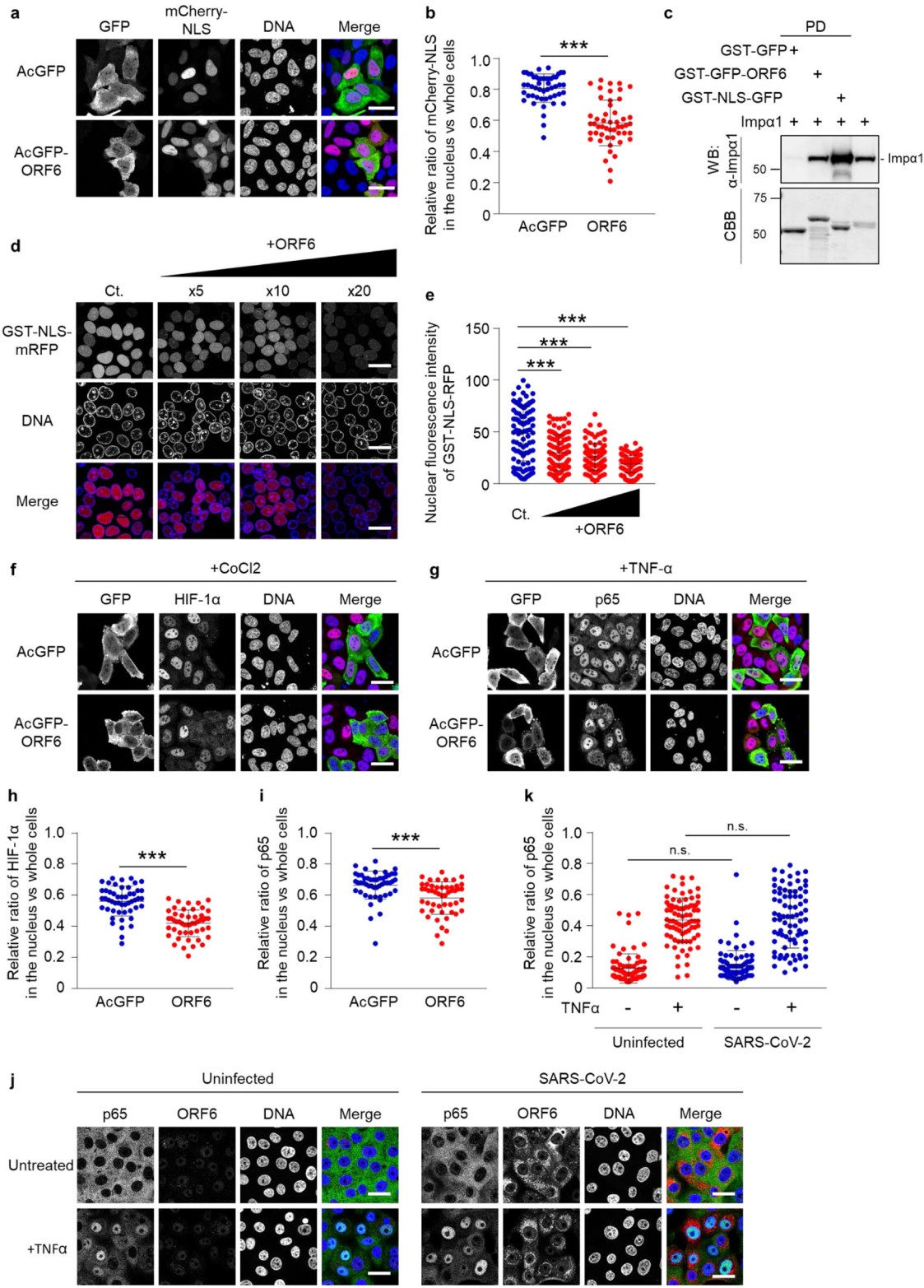

that ORF6 affects the importin α/β1 pathway, when it exists in large quantities in cells.

To further investigate the effects of ORF6 on other important signaling pathways mediated by importin α/β1 other than STAT1, we focused on the following signaling molecules, hypoxia inducible factor 1α (HIF-1α) and Nuclear factor-kappa B (NF-κB) component p65/RelA, since they have been shown to be transported into the nucleus by several importin α proteins

including importin α1[35–37]. First, HeLa cells were transfected with AcGFP or AcGFP-ORF6, and then treated with 200 µM cobalt chloride (CoCl2) for 5 h to induce the nuclear accumulation of HIF-1α. In addition, the nuclear migration of NF-κB p65 was investigated using AcGFP- or AcGFP-ORF6-transfected cells treated with 20 ng/mL tumor necrosis factor-α (TNF-α) for 30 min. As a result, the nuclear accumulations of both HIF-1α (Fig. 6f, h) and NF-κB p65 (Fig. 6g, i) were slightly but

**Fig. 6 ORF6 modestly disrupts the importin α/β1 pathway. a** Subcellular localization of mCherry-NLS (red) in HeLa cells transfected with AcGFP or AcGFP-ORF6 (green). DAPI was used to stain the DNA (blue). Scale bars: 30 μm. **b** The graph represents the relative fluorescence values of the nucleus compared to those of the whole cells in **a**. Signal intensities of total 50 nuclei from two independent experiments were measured. ***$P < 0.001$, two-tailed Student's t-test. error bars represent SD. **c** Importin α1 (Impα1) was incubated with GST-GFP, GST-GFP-ORF6, or GST-NLS-GFP immobilized on GST-beads for 1 h, and then a pulled-down protein was collected (PD). The importin α1 proteins were detected using the anti-importin α1 antibody (WB). The bottom panel represents the proteins bound to the beads and stained with CBB. The right lane was loaded importin α1 as an input protein (1/30,000 dilution of the reaction). Values are kDa. **d** An in vitro semi-intact nuclear transport assay was performed to measure the nuclear import of GST-NLS-mRFP in the presence of AcGFP-ORF6. Digitonin-permeabilized HeLa cells were incubated with GST-NLS-mRFP, importin α1, importin β1, RanGDP, p10/NTF2, GTP, and ATP regeneration system. The reaction mixture was added 5×, 10×, or 20× concentration of AcGFP-ORF6 compared to that of the NLS-substrate. After incubation for 30 min, the mRFP signals were detected using a fluorescence microscope. DAPI was used to stain the DNA. Scale bars: 30 μm. **e** The graph represents the nuclear fluorescence values of GST-NLS-mRFP in **d**. Signal intensities of total 100 nuclei were measured and analyzed using a one-way ANOVA (***$P < 0.001$). error bars represent SD. **f**, **g** Immunofluorescence of either HIF-1α **f** or NF-κB p65 **g** in HeLa cells transfected with AcGFP or AcGFP-ORF6 following CoCl2 treatment or TNF-α stimulation, respectively. Anti-GFP, anti-HIF-1α or anti-p65 antibodies were used for detection of AcGFP (green), HIF-1α (red), or p65 (red), respectively. DAPI was used to stain the DNA (blue). Scale bars: 30 μm. **h**, **i** The graphs represent the relative fluorescence values of HIF-1α **h** or NF-κB p65 (**i**) in the nucleus compared to those of the entire cells based on in **f** or **g**, respectively. Signal intensities of total 50 nuclei from two independent experiments were measured. ***$P < 0.001$, two-tailed Student's t-test. error bars represent SD. **j** Immunofluorescence of NF-κB p65 in Vero-TMPRSS2 cells infected with SARS-CoV-2 following TNF-α stimulation. Endogenous p65 (green) and viral ORF6 (red) were detected using the specific antibodies. **k** The graph represents the relative fluorescence values of p65 in the nucleus compared to whole cells in **j**. Signal intensities of a total of 80 nuclei from two independent experiments were measured. Two-tailed Student's t-test. n.s.: not significant. error bars represent SD. Scale bars: 30 μm.

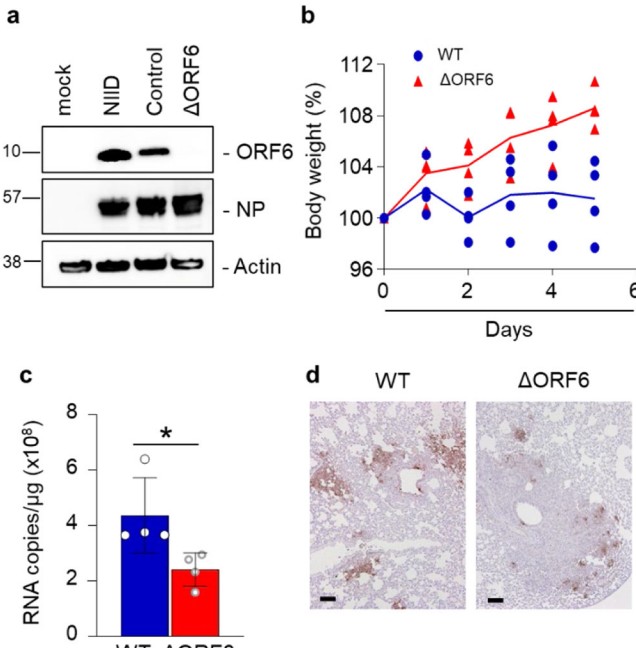

**Fig. 7 Deletion of ORF6 influences the pathogenicity in SARS-CoV-2 in vivo. a** Detection of ORF6 in recombinant SARS-CoV-2 control (Nluc-2A-ORF6) or ΔORF6. VeroE6/TMPRSS2 cells were infected with SARS-CoV-2 WT (Nluc-2A-ORF6) or ΔORF6 and the cell lysates were collected at 24 h post infection. Cell lysates were subjected to western blotting, and then detected the proteins using specific antibodies for ORF6, NP, or Actin. NIID: 2019-nCoV/Japan/TY/WK-521/2020 strain was isolated at the National Institute of Infectious Diseases. Values are kDa. **b** Percent body weight changes were calculated for all hamsters infected with SARS-CoV-2 WT (blue circle) or ΔORF6 (red triangle). Data are mean ± SD from four independent animals. **c** Viral RNA in lung homogenates from hamsters was quantified using qRT-PCR ($n = 4$). *$P < 0.05$, two-tailed Student's t-test. error bars represent SD. **d** Immunohistochemistry of SARS-CoV-2 antigen (NP protein) in lung lobes of hamster infected with SARS-CoV-2 WT or ΔORF6, respectively. Scale bars: 100 μm.

significantly suppressed in the ORF6-transfected cells. Notably, the inhibitory effects of ORF6 on the two importin α/β1-mediated signaling molecules were modest compared to that on STAT1. Moreover, the nuclear localization of p65 was not significantly prevented in SARS-CoV-2 infected cells (Fig. 6j, k). The results highly suggest that although the SARS-CoV-2 ORF6 potentially reduces the classical importin α/β1-mediated protein trafficking into the nucleus, the activity is more prominent for the STAT1-signaling pathway.

**ORF6 contributes to viral RNA replication and pathogenicity in vivo.** To further examine the roles of ORF6 in the viral life cycle of SARS-CoV-2, the viral genome encoding ORF6 was replaced by that of NanoLuc, and a SARS-CoV-2 variant that does not express ORF6 (SARS-CoV-2/ΔORF6) was generated using the circular polymerase extension reaction (CPER). As a parental virus, a recombinant virus expressing ORF6 fused with NanoLuc (NLuc) and Porcine teschovirus 2 A peptide (SARS-CoV-2/NLuc2AORF6) was generated (Supplementary Fig. 7a, b). The deletion of ORF6 in SARS-CoV-2/ΔORF6 was confirmed by western blotting along with the wild virus from NIID (Fig. 7a). To assess the viral growth, the recombinant viruses were inoculated at a multiplicity of infection (MOI) = 0.1 in Huh7-ACE2 cells and Vero-TMPRSS2 cells, respectively, and then the culture supernatants were collected at 6, 12, and 24 h post infection to determine the viral RNA and the viral titer. The results showed no alternation in the viral replication in both of Huh7-ACE2 cells (Supplementary Fig. 7c, d) and Vero-TMPRSS2 cells (Supplementary Fig. 7e, f).

Next, to evaluate the function of ORF6 in vivo, the parental virus (WT SARS-CoV-2) or SARS-CoV-2/ΔORF6 was intranasally inoculated in 4-week-old hamsters. Hamsters infected with SARS-CoV-2/ΔORF6 showed significant weight gain 5 days post-infection, whereas those infected with the WT virus showed no change in the body weight (Fig. 7b). In addition, 5 days post-infection, viral RNA was significantly reduced in the lung cells infected with SARS-CoV-2/ΔORF6 compared to those infected with the WT virus (Fig. 7c). Immunohistological analyses revealed that the viral nucleoprotein (NP) was expressed at lower levels in the lung cells infected with SARS-CoV-2/ΔORF6 than in those infected with the WT virus (Fig. 7d). The data indicate that ORF6 acts as a virulence factor in the pathogenesis of COVID-19.

## Discussion

In the present study, we report that ORF6 acts on the nucleo-cytoplasmic signaling via two distinct ways. That is, first, ORF6 inhibits the nuclear translocation of one of the key signaling molecules for COVID-19, STAT1, through its direct binding to antagonize the IFN signaling. Second, ORF6 has potential to impair the nuclear transportation of cNLS-containing cargos including signaling molecules such as HIF-1α and NF-κB p65, through affecting the function of importin α.

A previous study reported that SARS-CoV ORF6 tethered KPNA2 (importin α1 in this study), but not KPNA1 (importin α5), to the ER, and, as a result, sequestered importin β1 into the ER/Golgi segment through the interaction with KPNA2, resulting in the inhibition of PY-STAT1 nuclear import[7]. Since the nuclear transport of PY-STAT1 is known to be mediated by importin α5[25,26], and we demonstrate here that importin α1 can enter the nucleus even in the presence of ORF6 under oxidative stress conditions, it is unlikely that SARS-CoV-2 ORF6 tethers importin α1 to the ER to cause the nuclear exclusion of STAT1.

In contrast, we found that the Flag-STAT1 was more abundantly localized in the cytoplasm in the absence of IFN stimulation in the ORF6-transfected cells than in the control cells. It has previously been demonstrated that unphosphorylated STAT1 shuttles between the nucleus and the cytoplasm, and this shuttling might play an important role in regulating the expression of IFN stimulated genes[14,38–40]. Moreover, in this study, we found that the bacterially purified STAT1 proteins, which are not phosphorylated, bind to ORF6. Since the phosphorylation of STAT1 has been shown to be unaffected by ORF6 upon the IFN stimulation[8,11], the interaction between ORF6 and STAT1 might occur in a phosphorylation-independent manner. Collectively, we propose a scenario that the nuclear exclusion of STAT1 is caused by the direct binding with ORF6, independent of importin α proteins.

Conversely, we observed that the subcellular localization of all importin α subtypes were altered in ORF6-transfected cells, suggesting that ORF6 directly or indirectly influences the importin α/β1-mediated nuclear transport pathways. Consistently, we further found that the nuclear accumulation of the mCherry-NLS substrate was significantly reduced in the ORF6-transfected cells. In addition, using the semi-intact nuclear transport assay, the addition of excess amounts of ORF6 inhibits the nuclear transportation of GST-NLS-mRFP. Furthermore, ORF6 negatively regulates the nuclear import of HIF-1α and NF-κB p65, which have already been shown to be mediated by importin α proteins[35–37].

Recently, it has been shown that the specific interaction of ORF6 with NPC components, Nup98 and RAE1, might disrupt the nuclear trafficking of a broad range of proteins, in particular for the host mRNA export systems[10,29,41]. Indeed, our data revealed that the nuclear distribution of CAS, a nuclear exporter of importin α, was enhanced in the ORF6-transfected cells, suggesting that the nuclear export system may be affected considerably in the presence of ORF6. Conversely, we could not identify the dramatic suppression of p65 in SARS-CoV-2-infected cells. Since several recent papers have reported that viral components, such as Nsp5 or ORF7a, enhance cytokine expression through activating the NF-κB signaling pathway[42,43], the inhibitory effect of ORF6 may be counteracted by the other components. Overall, at least in the classical nuclear import pathway, the inhibitory effects of ORF6 could be limited, in contrast to the specificity for the STAT1-signaling pathway. Since the nucleo-cytoplasmic trafficking is vital for cell survival, SARS-CoV-2, therefore, should evade antiviral immune signaling without affecting cell survival to develop COVID-19.

Our microinjection analysis showed that only the C-terminal sequence of ORF6 could inhibit the nuclear localization of STAT1, whereas the nuclear exclusion of STAT1 was not likely completely achieved in the full-length ORF6-expressing cells. Interestingly, the GST-GFP-fused M0, M1, and M2 proteins all migrated into the nucleus. In contrast, the GST-M3-GFP protein localized only to the cytoplasm, like that with the control GST-GFP protein. This indicates that the C-terminal region of ORF6, in particular residues 56–61, has a potential as an NLS, but it is different from a classical NLS sequence based on a basic amino acid cluster. Notably, the M2 mutant lost its inhibitory effect on STAT1 nuclear localization. Specifically, ORF6 separately influences the STAT1-signal pathway and the importin α-mediated pathway. This is supported by our finding that although ORF6 suppressed the nuclear trafficking of the classical NLS-containing cargos through binding with importin α, the activity was more prominent with respect to STAT1 signaling. Together with the finding that the arginine substitution at methionine residue 58 of ORF6, which is deficient in Nup98 binding, abolishes its IFN antagonistic function[10], further studies are required to understand how ORF6 coordinates binding with either importin α, Nup98, or STAT1 itself through its C-terminal region to achieve the nuclear exclusion of STAT1. Understanding the effects of SARS-CoV-2 proteins on the nucleocytoplasmic trafficking system might facilitate the development of novel COVID-19 therapeutics.

## Materials and methods

**Animal care, and the production of monoclonal antibody**. All animal experiments using the SARS-CoV-2 virus were performed in biosafety level 3 (ABSL3) facilities at the Research Institute for Microbial Diseases, Osaka University. The animal experiments and the study protocol were approved by the Institutional Committee of Laboratory Animal Experimentation of the Research Institute for Microbial Diseases, Osaka University (R02-08-0). Throughout the study, we focused on minimizing animal suffering and reducing the number of animals used in the experiments. Four week-old male Syrian hamsters were purchased from SLC (Shizuoka, Japan).

Experimental procedures for production of monoclonal antibody were approved by the CEC Animal Care and Use Committee (permission number: CMJ-044) and performed according to CEC Animal Experimentation Regulations. A rat monoclonal antibody that specifically recognized the SARS-CoV-2 ORF6 protein was generated using the rat medial iliac lymph node method[44]. An 8-week-old female WKY rat was injected with 100 µL of emulsions containing ORF6 peptide (CEEQPMEID)-conjugated KLH and Freund's complete adjuvant into the rear footpads. Seventeen days after the first immunization, an additional immunization of SARS-CoV-2 ORF6 peptide-KLH was administered without an adjuvant into the tail base of the rat. Four days after the second immunization, cells from the iliac lymph nodes of the immunized rat were fused with mouse myeloma Sp2/0-Ag14 cells at a ratio of 5:1 in 50% polyethylene glycol. The resulting hybridoma cells were plated onto 96-well plates and cultured in HAT selection medium (Hybridoma-SFM [Life Technologies, Grand Island, CA, USA]; 10% FBS; 1 ng/mL mouse IL-6; 100 µM hypoxanthine [Sigma-Aldrich, St. Louis, MO, USA]; 0.4 µM aminopterin [Sigma-Aldrich]; and 16 µM thymidine [WAKO, Osaka, Japan]). The SARS-CoV-2 ORF6-specific antibody was screened using ELISA, western blotting, and immunostaining of hybridoma supernatants. Finally, hybridoma clone producing the monoclonal antibody, later named 8B10, was selected. Using a rat isotyping kit the MAb 8B10 was found to be an IgG 1 (k) antibody subtype. The monoclonal antibody against SARS-CoV-2 NP (3A9 clone) was generated by Cell Engineering Corporation (Osaka, Japan). Western blotting for the protein in cells infected with different viral strains, which were obtained from the National Institute of Infectious Diseases (NIID) in Japan, Hong Kong (HK)/VM20001061, USA-CA2, Germany/BavPat1, New York (NY)-PV09197, NY-PV08410, and NY-PV08449 (Supplementary Fig. 8a). Indirect immunofluorescence images for the SARS-CoV-2-infected VeroE6/TMPRSS2 cells, which is consistent with the previous observation in SARS-CoV-infected Vero E6 cells[45], are represented in Supplementary Fig. 8b.

**Viruses**. The SARS-CoV-2 (2019-nCoV/Japan/TY/WK-521/2020) strain was isolated at the National Institute of Infectious Diseases (NIID). The Germany/Bav-Pat1/2020, USA-CA2/20200 (USA-CA2), NY-PV08410/2020, HK/VM20001061, NY-PV08449/2020, and NY-PV09197/2020 strains were obtained from BEI Resources (Manassas, VA, USA). The different strains of SARS-CoV-2 were used to infect VeroE6/TMPRSS2 cells cultured at 37 °C with 5% CO$_2$ in DMEM (WAKO, Osaka, Japan) containing 10% fetal bovine serum (FBS; Gibco, Grand Island, NY, USA), and penicillin/streptomycin (100 U/mL, Invitrogen, Carlsbad, CA, USA). The viral stock was generated by infecting VeroE6/TMPRSS2 cells at a MOI of 0.1. The viral supernatant was harvested two days post infection and the viral titer was determined using plaque assay.

**Plasmid construction for mammalian expression**. The AcGFP and HA were amplified and cloned into pCAGGS vector designed as pCAG AcGFP-HA. The cDNA of ORF6 was obtained from Vero-TMPRSS2 cells infected with SARS-CoV-2. The wild type ORF6, ORF6-M1, ORF6-M2, and ORF6-M3 were amplified and cloned into pCAG AcGFP-HA designed as pCAG AcGFP-ORF6-HA, pCAG AcGFP-ORF6M1-HA, pCAG AcGFP-ORF6M2-HA, and pCAG AcGFP-ORF6M3-HA, respectively. The ORF6Δ9 was constructed using a splicing method achieved by overlap extension (ORF6Δ9-N and -C). The primers used throughout the study are described in Supplemental Table 1. All cDNAs were amplified using polymerase chain reaction (PCR) and the Tks Gflex DNA Polymerase (Takara Bio., Shiga, Japan). The amplified cDNAs were cloned into the indicated plasmids using an In-Fusion HD cloning kit (Clontech, Mountain View, CA, USA). The sequences of all plasmids were confirmed by Eurofins Genomics (Tokyo, Japan).

Full-length STAT1 was amplified from a previously subcloned plasmid[46] using the primers listed in Supplemental Table 1. The PCR products were cloned into a pcDNA5/FRT/3xFLAG expression vector[47]. Human importin αs including importin α1 (KPNA2) and importin α5 (KPNA1) were cloned into a pcDNA5/FRT/FLAG expression vector, as previously described[47]. For constructs encoding the SV40T antigen NLS (NLS: PKKKRKVED), the relevant oligonucleotides (Supplemental Table 1) were ligated into the pmCherry-C1 vector (Clontech).

The pISRE-TA-Luc plasmid was constructed as previously described[48]. The pGAS-TA-Luc plasmid was constructed as follows; the STAT1 sequence in the pGF1-STAT1plasmid (SBI-TR015PA-1-10, System Biosciences, LLC, Palo Alto, CA, USA) was cut off by EcoRI and Spe1, and then subcloned into the pGL4.10 plasmid (Promega, Madison, WI, USA).

**Plasmid constructions for bacterially expressed recombinant proteins**. The cDNAs of full-length ORF6 and the C-terminal mutant in which the amino acids 49–61 were deleted (ORF6ΔC) were amplified using the specific primers described in Supplemental Table 1. The PCR products were cloned into a pGEX6P2 vector (Clontech), which was subcloned into the GFP gene at the N-terminus[49]. Construct integrity was confirmed by DNA sequencing. For constructs encoding the C-terminus of ORF6 (M0) and its alanine replacing mutants (M1, M2, and M3), the relevant oligonucleotides (Supplemental Table 1) were annealed and ligated into a pGEX2T-GFP vector, which contained the GFP gene at the multicloning site; thus, producing the pGEX2T-M0-GFP, pGEX2T-M1-GFP, pGEX2T-M2-GFP, and pGEX2T-M2-GFP vectors. The plasmid pGEX6P2/hSTAT1 was subcloned from the pcDNA5/FRT/3xFLAG expression vector. The plasmids pGEX6P3/flag-human-importin α1 and pGEX6P3/flag-human-importin α5 were obtained as previously described[25,47]. The relevant oligonucleotides of SV40T NLS were ligated to the pGEX2T vector containing the monomeric RFP (mRFP) gene at the multicloning site; thus, producing the pGEX2T-NLS-mRFP vector.

**Purification of bacterially expressed recombinant proteins**. Purification of bacterially expressed recombinant proteins was performed as previously described[47,50]. Cleavage of the GST tag to induce cleaved fusion proteins was performed using PreScission protease (10 U/mg of the fusion protein, GE Healthcare, Uppsala, Sweden) or thrombin protease (10 U/mg of fusion protein, Sigma-Aldrich, Germany). Importin β1, p10/NTF2, and GDP-bound Ran were purified as previously described[47,50].

**Solution binding assay using recombinant proteins**. Solution binding assay using bacterially produced recombinant proteins was performed as previously described[47,50]. Bacterially produced FLAG-h-importins α1 and α5 or Flag-h-STAT1 recombinant proteins (100 pmol each) were incubated with GST-GFP, GST-GFP-ORF6, GST-M0-GFP, or GST-NLS-GFP immobilized on glutathione-Sepharose 4B beads (GST-beads, GE Healthcare, Tokyo, Japan), in 200 μL of transport buffer (TB; 20 mM HEPES, pH 7.3, 110 mM potassium acetate, 2 mM magnesium acetate, 1 mM EGTA, 1 mM DTT, 500 μM PMSF, and 1 μg/mL each of aprotinin, pepstatin, and leupeptin) with 0.1% Triton X-100 at 4 °C for 1 h. After the beads were washed five times with TB containing 0.1% Triton X-100, they were suspended in SDS-PAGE loading buffer (0.375M-Tris-HCl, pH 6.8, 0.03(w/v)%-BPB, glycerol, anion surface acting agent, and reducing agent, Nacalai Tesque, Kyoto, Japan).

**Antibodies**. The following primary antibodies were used in the present study: Phospho-STAT1 (Tyr701) (#9167 [58D6], Cell Signaling Technology (CST) Inc., Danvers, MA, USA), STAT1 (#9172, CST), importin α1/KPNA2 (ab84440, Abcam, Cambridge, MA, USA; #610486, Anti-Karyopherin α (Rch1), BD Transduction Lab., San Jose, CA, USA), HIF-1α (ab51608 [EP1215Y], Abcam), NF-κB p65 (#8242 [D14E12], CST), importin β1 (ab2811 [3E9], Abcam), CAS (ab96755, Abcam), Lamin A/C (sc-6215, Santa Cruz, Dallas, TX, USA), Flag (M2 [F1804], Sigma-Aldrich), GFP (A-11122, rabbit, Thermo Fisher Scientific, Waltham, MA, USA), GFP (M048-3, mouse, MBL, Nagoya, Japan), NP (3A9, mouse mAb, Cell Engineering Co., Osaka, Japan), Actin (A2228, Sigma-Aldrich), and HA (MMS-101R, Biolegend, San Diego, CA, USA).

Horseradish peroxidase (HRP)-conjugated anti-rabbit (#111-035-003), anti-mouse (#115-035-003), or anti-rat (#112-035-003) secondary antibodies (Jackson ImmunoResearch Inc. West Grove, PA, USA) were used for western blotting. The secondary antibodies used for indirect immunofluorescence were as follows: Alexa

Fluor Plus 488 conjugated anti-rabbit (A32731) or anti-mouse (A32723), and Alexa Fluor 594 conjugated anti-rabbit (A21207) or anti-mouse (A21203) (Invitrogen).

**Cell culture and transfection**. HeLa cells (ATCC), HEK293 cells (NIBIOHN), Huh7 cells (National institute of infectious diseases in Japan.), Huh7-ACE2 which were generated by infection with lentivirus expressing human ACE2, VeroE6/TMPRSS2 cells (NIBIOHN, JCRB1819), and Vero E6 replicon stable cells[51] were cultured in Dulbecco's modified Eagle's medium (DMEM; Invitrogen), containing 10% FBS (#10270, Gibco) at 37 °C in 5% $CO_2$. The cells were plated onto 18 × 18 mm coverslips (Menzel-Glaser, Braunschweig, Germany) in 35-mm dishes for immunofluorescence or 60-mm dishes (IWAKI, Tokyo, Japan) for qRT-PCR 2 days prior to transfection. The transfections were performed using Lipofectamine 2000 DNA Transfection Reagent (Thermo Fisher Scientific) or the TransIT-LT1 Transfection Reagent (Mirus, Madison, WI, USA) following manufacturer's instructions.

**Indirect immunofluorescence**. HeLa cells were cultured on 18 × 18 mm coverslips (Matsunami, Osaka, Japan) in 35-mm dishes (IWAKI) and incubated for 48 h at 37 °C in 5% $CO_2$. The reagents used for indirect immunofluorescence were as follows; IFN-β and IFN-γ (final conc. was 50 ng/mL for 30 min; Miltenyi Biotec, Bergisch Gladbach, Germany), TNF-α (final conc. was 20 ng/mL for 30 min; Miltenyi Biotec), hydrogen peroxide ($H_2O_2$, final conc. was 200 μM for 1 h), and Cobalt(II) chloride hexahydrate (CoCl2, final conc. was 200 μM for 5 h; C8661, Sigma-Aldrich). Following fixation with 3.7% formaldehyde in PBS for 15 min, cells were treated with 0.1% Triton X-100 in PBS for 5 min and then blocked in PBS containing 3% skim milk for 30 min. For the anti-Phospho-STAT1 antibody (58D6, Rabbit mAb, #9167, CST), cells were permeabilized with 100% methanol at -20 °C for 20 min and then blocked in 3% skim milk in PBS. Cells were incubated with primary antibodies (1:200) with 3% skim milk in PBS overnight at 4 °C. The following day, the cells were incubated with the Alexa-Fluor-488 plus- or Alexa-Fluor-594-conjugated secondary antibodies (Invitrogen). Nuclei were counterstained with DAPI (1:5,000 in PBS, Dojindo Laboratories, Kumamoto, Japan) for 20 min at 25 °C. The coverslips with fixed cells were mounted on glass slides using ProLong Gold Antifade (#36930, Invitrogen). Cells were examined under a confocal microscope (Leica TCS SP8 II; Leica Microsystems, Wetzlar, Germany). Using Leica Application Suite X, cells only expressing AcGFP were extracted and then the fluorescence intensities were identified at a region of interest in the nucleus, as well as in the whole cells. The relative fluorescence intensity values in the nuclei against the whole cells were calculated.

**Western blotting**. Western blotting was performed as previously described[52]. The membranes were incubated with primary antibodies (dilutions ranging from 1:1000 to 1:2000) diluted in Can Get Signal Immunoreaction Enhancer Solution 1 (TOYOBO, Osaka, Japan) overnight at 4 °C. The used HRP-conjugated secondary antibodies (dilutions ranging from 1:2000 for mammalian expression to 1:10,000 for bacterially purified recombinant proteins) were diluted in Can Get Signal Immunoreaction Enhancer Solution 2 (TOYOBO) at 25 °C for 1 h.

**Immunoprecipitation**. HEK293 cells ($0.5–2 \times 10^7$) were suspended with 1 mL RIPA buffer (Nacalai Tesque) and lysed by successive passage through 26-gauge needles (3 times each). The samples were kept on ice for 20 min and then insoluble material was removed by centrifugation at $13,000 \times g$ at 4 °C for 15 min. The supernatant was precleared with 20 μL of Dynabeads (M-280 anti-mouse IgG, Invitrogen) at 4 °C for 1 h, followed by incubation of the pre-cleared cell lysates with 20 μL Dynabeads and 2 μg of a primary antibody specific for flag (M2 [F1804], Sigma-Aldrich) at 4 °C for 2 h. The beads were washed five times with 1 mL Lysis buffer and bound proteins eluted with the addition of SDS-PAGE loading buffer (Nacalai Tesque).

**RNA purification and quantitative RT-PCR (qRT-PCR)**. For IP-10, total RNA was isolated using ReliaPrep™ RNA Tissue Miniprep System (Promega) according to the manufacturer's instructions. One microgram of total RNA and the Prime-Script RT reagent kit (Takara Bio.) were used to perform the first-strand cDNA synthesis. The PCR reaction was performed as previously described[52]. The PCR primers including those of β-actin are described in Supplemental Table 2.

For detection of N2 in SARS-CoV-2, total RNA of Huh7-ACE2 or lung homogenates were isolated using ISOGENE II (Nippon Gene, Toyama, Japan). Real-time RT-PCR was performed with the Power SYBR Green RNA-to-CT 1-Step Kit (Applied Biosystems, Foster City, CA, USA) using an AriaMx Real-Time PCR system (Agilent, Santa Clara, CA, USA). The relative quantification of the target mRNA levels was performed using the $2^{-\Delta\Delta CT}$ method. β-actin was used as the housekeeping gene. The primers used are described in Supplemental Table 2.

**Quantitative RT-PCR of viral RNA in the supernatant**. The amount of RNA copies in the culture medium was determined using a qRT-PCR assay as previously described with slight modifications[53]. Briefly, 5 μL of culture supernatants were mixed with 5 μL of 2× RNA lysis buffer (2% Triton X-100, 50 mM KCl, 100 mM Tris-HCl [pH 7.4], 40% glycerol, 0.4 U/μL of Superase•IN [Thermo Fisher

Scientific]) and incubated at 25 °C for 10 min. Next, 90 µL of RNase free water were added to the mix. A volume of 2.5 µL of the diluted sample was added to 17.5 µL of reaction mix. Real-time RT-PCR was performed using the Power SYBR Green RNA-to-CT 1-Step Kit (Applied Biosystems) and an AriaMx Real-Time PCR system (Agilent).

**Plaque formation assay**. Vero-TMPRSS2 were seeded into 24-well plates (80,000 cells/well) at 37 °C in 5% $CO_2$ for overnight. The supernatants were serially diluted using inoculated medium and incubated for 2 h. Next, the culture medium was removed, fresh medium containing 1% methylcellulose (1.5 mL) was added, and the cells were cultured for 3 more days. Lastly, the cells were fixed with 4% paraformaldehyde in PBS (Nacalai Tesque) and the plaques were visualized by using a Giemsa's azur-eosin-methylene blue solution (#109204, Merck Millipore, Darmstadt, Germany).

**Syrian hamster model of SARS-CoV-2 infection**. Syrian hamsters were anaesthetized with isoflurane and challenged with $1.0 \times 10^6$ PFU (in 60 µL) SARS-CoV-2 via intranasal routes. The body weight was monitored daily for 5 d. Five days postinfection, all animals were euthanized, and the lungs were collected for histopathological examinations and qRT-PCR.

**Immunohistochemistry**. Lung tissues were fixed with 10% neutral buffered formalin and embedded in paraffin. For immunohistochemical staining, 2 µm thick sections were immersed in citrate buffer (pH 6.0) and heated for 20 min with a pressure cooker. Endogenous peroxidase was inactivated by immersion in 3% $H_2O_2$ in PBS. After treatment with 5% skim milk in PBS for 30 min at 25 °C, the sections were incubated with mouse anti-NP antibody (1:500, clone 3A9). EnVision+ system-HRP-labeled polymer anti-mouse secondary antibody (Dako, Carpinteria, CA, USA) was used. Lastly, the sections were counterstained with hematoxylin and the positive signals were visualized using the peroxidase–diaminobenzidine reaction.

**Construction of SARS-CoV-2 replicon DNA**. SARS-CoV-2 replicon vector, pBAC-SCoV2-Rep, was generated using the CPER reaction as previously described[54], with some modifications[51]. Briefly, seven DNA fragments covering the SARS-CoV-2 genome (excluding the region spanning the S gene to ORF8 gene) were amplified using PCR, and subcloned into a pCR-Blunt vector (Invitrogen). The DNA fragments containing cytomegalovirus (CMV) promoter, a 25-nucleotide synthetic poly(A), a hepatitis delta ribozyme, as well as a bovine growth hormone (BGH) termination, and a polyadenylation sequences (the lightly shaded region in Fig. 2d) were amplified using a conventional overlap extension PCR, and subcloned into the NotI sites of pSMART BAC vector (Lucigen, Middelton, WI, USA). The luciferase reporter vector pGL4 was used as the template for PCR amplification of Renilla luciferase gene. For CPER reaction, nine DNA fragments that contain approximately 40-bp overlapping ends for two neighboring fragments were amplified by PCR using the aforementioned plasmids. Next, the PCR fragments were mixed equimolarly (0.1 pmol each) and subjected to CPER reaction using the PrimeSTAR GXL DNA polymerase (Takara Bio.). The CPER product was extracted using phenol-chloroform, followed by ethanol precipitation, resolved in TE buffer, and transformed into the BAC-Optimized Replicator v2.0 Electrocompetent Cells (Lucigen). The replicon vector was maxipreped using a NucleoBond Xtra BAC kit (Takara Bio.).

**Transient replicon assay**. Huh7 cells were seeded onto 24 well plates, and then transfected with the replicon vector together with AcGFP or AcGFP-ORF6 vector, using TransIT-LT1. Cells were harvested 24 h post-DNA transfection, and luciferase activity was determined using a Renilla luciferase assay system (Promega). For the RNA interference experiment, small interfering RNAs (siRNA) targeting STAT1 (#1, SASI_Hs01_00108158; #2, SASI_Hs02_00364253) and control siRNA (MISSION siRNA Universal Negative Control #1 (Cat No. SIC001)) were purchased from Sigma-Aldrich. Huh7 cells were transfected with siRNA using Lipofectamine RNAiMAX (Invitrogen), and the downregulation of the target protein expression was analyzed by western blotting. For the transient replicon assay, the cells were seeded onto 24 well plates and transfected with siRNA overnight. Subsequently, the cells were transfected with the replicon vector, together with AcGFP or AcGFP-ORF6 vector, and harvested 24 h post-DNA transfection.

**Generation of SARS-CoV-2 recombinant virus**. SARS-CoV-2 recombinants were generated by CPER reaction as previously described[55] with some modifications. Briefly, 14 SARS-CoV-2 (2019-nCoV/Japan/TY/WK-521/2020) cDNA fragments (#1-#13) were amplified using PCR and subcloned into a pBlueScript KS(+) vector. The primers used are described in Supplemental Table 3. The DNA fragments containing CMV promoter, a 25-nucleotide synthetic poly(A), hepatitis delta ribozyme and BGH termination and, polyadenylation sequences (#14) were synthesized by Integrated DNA Technologies (Coralville, IA, USA), and subcloned into a pBlueScript KS(+) vector. To generate a reporter SARS-CoV-2 virus, we inserted a NanoLuc (NLuc) gene and 2 A peptide into the ORF6 sequence of fragment #12 (SARS-CoV-2/NLuc2AORF6). To generate an ORF6 deficient SARS-

CoV-2 virus, ORF6 gene was replaced with an NLuc gene (SARS-CoV-2/ΔORF6). For CPER reaction, 14 DNA fragments that contain approximately 40- to 60-bp overlapping ends for two neighboring fragments were amplified using PCR from the subcloned plasmids. Next, the PCR fragments were mixed equimolarly (0.1 pmol each) and subjected to CPER reaction using the PrimeSTAR GXL DNA polymerase (Takara Bio.). The cycling condition included an initial 2 min of denaturation at 98 °C; 35 cycles of 10 s at 98 °C, 15 s at 55 °C, and 15 min at 68 °C; and a final elongation period of 15 min at 68 °C. The half of CPER product was transfected into IFNAR1-deficient HEK293 cells TransIT-LT1 transfection reagents (Mirus), according to the manufacturer's instructions. ACE2 and TMPRSS2 receptors were induced in HEK293-3P6C33 cells using tetracycline. At 24 h posttransfection, the culture medium was replaced with DMEM containing 2% FBS and doxycycline hydrochloride (1 µg/mL). At 7–10 days post transfection, the culture medium containing progeny viruses (P0 virus) were passaged and amplified in VeroE6/TMPRSS2 cells.

**Luciferase assay**. Huh7 cells were seeded into a 24-well plate and incubated at 37 °C for 24 h. The cells were transfected with either pISRE-TA-Luc or pGAS-TA-Luc, and pRL-TK (Promega), as well as pCAG AcGFP-HA or pCAG AcGFP-ORF6-HA using TransIT-LT1 reagents (Mirus), according to the manufacturer's instructions. The cells were incubated for 24 h after transfection and treated with IFN-γ (50 ng/mL) for 12 h.

VeroE6/TMPRSS2 cells were seeded into a 24-well plate and incubated at 37 °C for 24 h. The cells were transfected with either pISRE-TA-Luc or pGAS-TA-Luc and pRL-TK (Promega). After 24 h, the cells were infected with SARS-CoV-2 and treated with IFN-γ (50 ng/mL) for 12 h.

Luciferase activity was detected using the Dual-Luciferase Reporter Assay System (Promega) according to the manufacturer's instructions. Relative luciferase values were calculated based on the firefly/Renilla luciferase values of AcGFP.

**Semi-intact nuclear transport assay**. A digitonin-permeabilized in vitro nuclear transport assay was performed as previously described[47]. The NLS substrate GST-NLS-mRFP (4 pmol) was used in 10 µL of reaction mixture, and the competitive substrate AcGFP-ORF6 was added to the assay with 20 pmol, 40 pmol, and 80 pmol, which represented 5×, 10×, or 20× the NLS-substrate dosages, respectively.

**Microinjection**. Huh7 cells were cultured in a 35-mm glass bottom dish (Matsunami), and either GST-GFP, GST-M0-GFP, GST-M1-GFP, GST-M2-GFP, and GST-M3-GFP recombinant proteins (final conc. 1 mg/mL) were microinjected into the cytoplasm with AlexaFluor555-conjugated antibody as an injection marker (final conc. 2 mg/mL; Thermo Fisher Scientific). The cells were incubated with or without IFN-γ (final conc. was 50 ng/mL) for 30 min.

**Statistics and reproducibility**. For most analyses, data are shown as the mean ± standard deviation (SD) from 2 to 4 independent experiments. Data were analyzed with Prism 7.0 software (GraphPad Software, La Jolla, CA). Statistical significance was evaluated by one-way Analysis of Variance (ANOVA) or two-way ANOVA for comparison of multiple groups, and the Student t test for two groups, *$P < 0.05$, **$P < 0.01$, ***$P < 0.001$.

**Reporting summary**. Further information on research design is available in the Nature Research Reporting Summary linked to this article.

## Data availability
All data generated or analyzed during this study are included in this published article, Supplementary Figs. 1–8 and Supplementary Tables 1–3. Source data underlying the graphs are provided in Supplementary Data 1. The original, unprocessed western blot/CBB gel images can be found at the end of Supplementary Information. The data that support the findings of this study are available from the corresponding author upon reasonable request.

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

## Acknowledgements
This work was funded by the Japan Agency for Medical Research and Development (AMED) [grant numbers 20fk0108263h0001, 20fk0108296s0101, and 20fk0108518h0001] to Y.M., T.S., T.Tanaka, and T.O.

## Author contributions
Conceptualization: Y.M. and T.O. Methodology: Y.M., T.S., T.Tanaka, Y.S., M.K., T.Tachibana, Y.K., Y.Y., and T.O. Investigation: Y.M., Y.I., T.S., T.Tanaka, Y.S., M.K., C.H., C.W., M.Otani, and T.O. Resources: Y.M., T.S., T.Tanaka, Y.S., K.M., T.Tachibana, Y.K., T.O., and M.Oka Writing—Original Draft: Y.M., Y.I., T.S., T.Tanaka, Y.S., M.K., T.Tachibana, Y.K., Y.Y., T.O., and M.Oka Writing—Review & Editing: Y.M., Y.I., T.S., T.Tanaka, Y.S., M.K., C.H., C.W., M.Otani, K.M., T.Tachibana, Y.K., Y.Y., T.O., and M.Oka Funding Acquisition: Y.M., T.S., T.Tanaka, and T.O Supervision: Y.M. and T.O.

## Competing interests
The authors declare no competing interests.
