## [Peer Review File · Communications Biology]

Reviewers' comments:

Reviewer #1 (Remarks to the Author):

This paper suggests that SARS-CoV-2 ORF6 inhibits importin α 1 to promote viral replication. Although it is potentially interesting and important, deficiencies in several areas were detected. Too many claims were made and the experimental support to some are weak. Some discrepancies were unexplained. Too many experiments were done in the irrelevant HeLa cells. Much netter evidence is required to support the conclusions of the paper.

1) Better evidence is required to support that reduced replication of Δ ORF6 virus is due to type I IFN induction, STAT1 or importin α 1. Mechanistic study should also be performed to elucidate how ORF6 interacts with importin α 1 and STAT1 to exert its function. Does it compete with importin α 1 for binding with STAT1. Co-immunoprecipitation should be performed with anti-ORF6, anti-importin α 1 and anti-STAT1.

2) It was also shown that ORF6 M2 and M3 mutants failed to prevent nuclear translocation of STAT1, although it has already been reported in Ref 8. Can these mutants bind with STAT1 or importin α 1? This should be tested experimentally.

3) It was claimed that nuclear exclusion of STAT1 by ORF6 is not due to the cytoplasmic tethering of importin α 1 (lines 243-44). This claim is weak as it was only shown that cytoplasmic localization of importin α 1 in the presence of ORF6 was rescued by treatment with hydrogen peroxide. Cytoplasmic retention of STAT1 or type I ISG expression should be examined.

4) Fig. 1: If ORF6's sole role is to suppress STAT1 to enhance viral replication, there should not be difference between WT and Δ ORF6 viruses in Vero cells since Vero cells are IFN-deficient. This experiments should be added. Viral kinetics of WT and Δ ORF6 were measured with a single time point (24hpi). More time points or a curve should be shown to see the impact of Δ ORF6 on viral replication (similar to Fig. 6J and K). As viral RNA may not reflect infectious titer in hamster model, TCID50 should be provided or plaque assay should be performed (Fig. 1F).

5) Fig. 2E and F: GAS luciferase is needed. Fig. 2F has a big SD as indicated by error bars and the experiment should be repeated.

6) Fig. 3E and lines 176-178: STAT1 KO or KD cells should be used.

7) Several lines of experiments should be performed to strengthen the interaction between ORF6 and importin α 1: a) Co-immunoprecipitation is required in addition to GST pull-down in Fig. 5D. b) GFP only control is required for Fig. 5E, 5F and S3C. c) Nuclear import of importin α 1 was unaffected by ORF6 (Fig. 5E and F). Whether ORF6 affects RanGTP/CAS/importin α 1 export should be tested (ORF6 reduces cytoplasmic CAS, an importin α 1 exporter (Fig. S3E)). d) The effect of ORF6 on endogenous importin α 1 is weak at best (Fig. S3A and S3B). It is required to evaluate importin α 1 distribution during IFN- γ treatment or infection e.g. with Δ ORF6 virus. e) Is it STAT1 dependent? (ORF6-M2 and -M3 can be used as suggested in point 2 above).

8) Importin α 1 knockout only had minimal effect on SARS-CoV-2 replication (Fig. 6J and K): a) Can a more pronounced effect be observed with Δ ORF6 virus? b) Importin α 3, α 4, α 6, and α 8 may have substituted the role of importin α 1 (Fig. S2). Chemical inhibitors (e.g. Ivermectin) might be used.

9) Subcellular fractionation experiment is required to support some critical staining data e.g. for STAT1, importin α 1 and α 5.

Reviewer #2 (Remarks to the Author):

Synopsis: This study examines the mechanism of SARS-CoV-2 ORF6 mediated inhibition of

nucleocytoplasmic transit in infected cells. As discussed in the introduction of the manuscript, this subject has been investigated previously by several groups for ORF6 of SARS-CoV and well as SARS-CoV-2, primarily with regard to nuclear localization of STAT1 in response to interferon stimulation. These earlier studies suggest that ORF6 prevents transport of phosphor-STAT1 to the nucleus through binding to importin $\alpha 1$ or the nuclear pore components Nup68-RAE1. The current study extends this line of inquiry. Initially, the authors find that the replication of SARS-CoV-2 lacking ORF6 is attenuated in vitro and in vivo. In agreement with previous data, they observe that expression of ORF6 inhibits STAT1 localization in the presence of IFN- β , as well as IFN- γ . Focusing on IFN- γ , they also show reduced expression of IP-10 in ORF6-expressing cells in response to IFN- γ treatment, as well as reduced expression of luciferase from an ISRE-driven reporter. They observed this effect to be dependent on the C-terminus of ORF6, in agreement with previous studies. Notably, they next demonstrate direct binding of ORF6 to STAT1, which occurs even in the absence of IFN- γ stimulation. Despite this novel observation, they also observe sequestration of importin $\alpha 1$ as well as other importins in the cytoplasm by ORF6. However, the localization of importins $\alpha 5$ and $\alpha 7$ was less affected by the presence of ORF6. They also show binding of recombinantly expressed GST-ORF6 and importin $\alpha 1$, but not $\alpha 5$. In order to examine the effect of ORF6 on the nuclear import of other proteins, the authors look at localization of HIF-1 α in response to CoCl $_2$, and NF- κ B p65 in response to TNF α . They observed a significant inhibition of nuclear localization of these proteins, although this is very hard to see in figure 6H. To examine the effect of importin $\alpha 1$ on SARS-CoV2 replication, the gene was knocked out Huh7-ACE2 cells, followed by infection. The knockout cells showed a very modest increase in virus replication, suggesting a role for importin $\alpha 1$ in suppressing replication.

From these data, the authors conclude that ORF6 functions to block STAT1 signaling via a direct interaction between these proteins. Further, ORF6 also appears to relocalize several importin proteins, but does not prevent their nuclear localization under conditions of oxidative stress. However, ORF6 expression does inhibit other aspects of nuclear localization, as shown for p65 and HIF1 α . Although the data are consistent with the authors' conclusions, several additional experiments would help to strengthen this study.

Comments/ concerns:

1. In figure 1D, the growth differences between the wt virus and Δ ORF6 should be shown for several times post infection. Viral load on additional days post infection should also be added to figure 1F. Are there significant differences in virus induced lung pathology in hamsters infected with Δ ORF6 or WT virus?
2. In the Luc-2a-ORF6 virus, is ORF6 expression the same as in true wt virus?
3. In fig 4E, could importin $\alpha 1$ be part of complex with STAT1/ ORF6? Is it possible that there is minimal stimulation of the IFN response via transfection? Is the STAT1 pulled down clearly not phosphorylated? (This is suggested in the discussion but not shown). Does the reciprocal pull down (IP: HA) also show this interaction?
4. Fig. 4 F and G: Does the ORF6 with the C-terminal deletion fail to interact with STAT1?
5. Fig. 6: Knockout of importin $\alpha 1$ / KPNA2 should be verified by western blot.
6. The effect of the KPNA2 knockout on virus replication is quite modest, at best. Does the KPNA2 knockout rescue the Δ ORF6 phenotype? This might be expected to show a greater effect? What is the effect of a STAT1 knockout on wt and Δ ORF6 viruses?
7. Fig 6h does not appear to support an effect on p65 nuclear localization – could this be shown another way? (e.g. ORF6 effect on NF- κ B responsive reporter, western blotting of nuclear vs. cytoplasmic fractions)?
8. Does SARS-CoV-2 infection affect NF- κ B or HIF-1 α signaling?

Minor points:

9. What is the methodology for the results in Fig. 2C and similar figures? Are only GFP+ cells counted?
10. Fig 2E: The ISRE responds directly to STAT1/2 heterodimers induced by type I interferon. An IFN γ responsive element (GAS) should be used in this experiment.
11. ORF6 appears to have an effect on importin $\alpha 5$, although to a lesser degree (Fig 5C), but text seems to imply there was no effect.

Reviewer #3 (Remarks to the Author):

It has been previously known that the ORF6 can inhibit activation of host innate immune responses via blocking the translocation of STAT1 (a key mediator of innate immunity) from the cytoplasm into the nucleus. Miyamoto and co-workers have now provided additional evidence to shed light on the underlying molecular mechanism. The authors demonstrate that SARS-CoV-2 can interact with STAT1 and withhold the mediator in the cytoplasm, thus preventing the subsequent activation of the innate immune system in the nucleus. Furthermore, ORF6 can selectively interact with an importin protein, KPNA2, to disrupt the trafficking of specific host proteins between the cytoplasm and nucleus. These findings suggest that SARS-CoV-2 can employ multiple strategies to dampen the impact of the front line of host defense. In sum, this manuscript highlights the fundamental role of ORF6 -- to antagonize the activation of host innate immune systems by modulating an innate immune signaling mediator as well as a cytoplasmic-nuclear trafficking component.

This manuscript with high-quality results is well written. In addition to the new knowledge, the authors may contribute invaluable reagents, i.e., the antibody against ORF6, infectious clone cDNAs, replicon systems, to the research community.

Nonetheless, there are a few specific comments that may be useful to strengthen the manuscript.

Specific comments:

Fig.3E:

The methodology for this section is incomplete – it is not clear how the pBAC-derived replicon was delivered into the cells and how the luciferase signal was obtained. Please include this in the manuscript.

Fig.2E, 2F, 3E: It would be easier to appreciate the actual differences by using raw luciferase signals (instead of normalized values). Please consider re-plotting the graphs using raw luciferase signals. Also, it is not clear how the authors computed the “relative luciferase values.” Please clarify in the text to make it easier for the audience to appreciate the findings fully.

Fig.5A & 5C and Fig. S3A & 3B: please explain why endogenous importin alpha 1 seems to predominantly localize in the cytoplasm but not the FLAG-tagged version of the protein that was ectopically expressed in cells. The FLAG-tagged version is pretty much present in the nucleus. If this discrepancy were valid, it would be fair to question whether FLAG-tagging could alter the subcellular localization of these importin proteins (i.e., Fig. 5A, 5B, and Fig. S2)

Fig. 6J and 6K: please provide evidence to show that KPNA2 is being CRISPR knocked out (e.g., WB or IFA; should be pretty straightforward as the authors have presented immunofluorescent imaging for KPNA2 in Fig. S3). Please include results for validating the knockout(s). How do you know that these are “on-target” knockouts? It is common to include a “cDNA addback” to see if the phenotype could be rescued. Please include the rescue experiment (should be straightforward as the KPNA2 cDNA was used in Fig. 5). Also, it is not clear if one sgRNA or three sgRNAs were used to edit KPNA2 (see line 276 vs. line 509). Please clarify.

Response to Reviewer 1

1) Better evidence is required to support that reduced replication of Δ ORF6 virus is due to type I IFN induction, STAT1 or importin α 1. Mechanistic study should also be performed to elucidate how ORF6 interacts with importin α 1 and STAT1 to exert its function. Does it compete with importin α 1 for binding with STAT1. Co-immunoprecipitation should be performed with anti-ORF6, anti-importin α 1 and anti-STAT1.

Answer 1

Following the reviewer's suggestion, we performed co-immunoprecipitation to detect the binding of ORF6 with importin α s or STAT1. Fig. 3E in the revised manuscript reveals that AcGFP-ORF6 was co-precipitated with Flag-STAT1 using anti-Flag antibody. Notably, the binding occurred with or without IFN stimulation, which was clearly showed by PY-STAT1. This supports our hypothesis that ORF6 directly interacts with STAT1 even in the absence of IFN stimulation.

Next, another co-IP dataset is presented in Fig. 4D in the revised manuscript to elucidate the binding status between ORF6 and importin α s. The sample precipitated with Flag-importin α 1 apparently contained AcGFP-ORF6 with or without the IFN stimulation. Conversely, binding with Flag-importin α 5 showed the IFN stimulation dependency. In the unstimulated condition (Untreated), AcGFP-ORF6 was certainly but faintly detected in the sample precipitated by Flag-importin α 5. This corresponds to the partial distribution change of importin α 5 showed in Fig. 4A and B. In contrast, the band intensity of AcGFP-ORF6 was clearly increased in response to the IFN stimulation, which was identified by PY-STAT (the right lane in Fig. 4D), which is consistent with a previous report that PY-STAT1 is recognized by importin α 5 (referred to as NPI-1 in the original paper), not importin α 1 (referred to as Rch1) ¹.

The results indicate that ORF6 binds to STAT1, in addition to importin α 1, with or without IFN stimulation. In addition, ORF6 does not compete with importin α 1 for binding with STAT1 because the interaction of ORF6 with importin α 1 was not affected by IFN stimulation. While PY-STAT1 is mainly recognized by importin α 5 to be transported into the nucleus, ORF6 may be a complex with importin α 5 through PY-STAT1 to inhibit the nuclear transport. The data support our hypothesis that nuclear exclusion of STAT occurred by the direct binding of ORF6 to STAT1, independent of the binding to importin α s. Thank you very much for the useful comments.

2) It was also shown that ORF6 M2 and M3 mutants failed to prevent nuclear translocation of STAT1, although it has already been reported in Ref 8. Can these mutants bind with STAT1 or importin α 1? This should be tested experimentally.

Answer 2

Following the reviewer's suggestion, we attempted to perform a co-immunoprecipitation assay for the mutants. Aligning the protein expression levels of ORF6 M2 and M3 mutants for a pull-down assay was very challenging. Therefore, we generated a new deletion mutant lacking the C-terminal sequence

(49-61 amino acids), as suggested by Reviwer#2, and then examined the binding with the STAT1 recombinant protein (Fig. 3H in the revised manuscript). Our data indicate that the C-terminus deletion reduced binding with STAT1 dramatically. We hope that the experimental data are satisfactory for the reviewer.

3) *It was claimed that nuclear exclusion of STAT1 by ORF6 is not due to the cytoplasmic tethering of importin α 1 (lines 243-44). This claim is weak as it was only shown that cytoplasmic localization of importin α 1 in the presence of ORF6 was rescued by treatment with hydrogen peroxide. Cytoplasmic retention of STAT1 or type I ISG expression should be examined.*

Answer 3

In response to the reviewer's concern, we examined the subcellular distribution of STAT1 under hydrogen peroxide. As shown in supplemental Fig. S5C and D, in the revised manuscript, Flag-STAT1 localized in the cytoplasm under the oxidative stress condition even though the distribution of importin α 1 was shifted to the nucleus by the same condition (Fig. 4E, F, and Fig. S5A, B in the revised manuscript). Such evidence clearly demonstrate that nuclear exclusion of STAT1 by ORF6 is not due to the cytoplasmic tethering of importin α 1. We appreciate the reviewer's input, which has improved the manuscript.

4) *Fig. 1: If ORF6's sole role is to suppress STAT1 to enhance viral replication, there should not be difference between WT and Δ ORF6 viruses in Vero cells since Vero cells are IFN-deficient. This experiments should be added. Viral kinetics of WT and Δ ORF6 were measured with a single time point (24hpi). More time points or a curve should be shown to see the impact of Δ ORF6 on viral replication (similar to Fig. 6J and K). As viral RNA may not reflect infectious titer in hamster model, TCID50 should be provided or plaque assay should be performed (Fig. 1F).*

Answer 4

Following the reviewer's critical suggestion, we performed a viral replication assay with several time points, in Vero-TMPRSS2 cells infected with WT and Δ ORF6 virus. However, in contrast to the original data, the new results did not replicate the reduction not only in Vero-TMPRSS2 cells but also in Huh7-ACE2 cells, even at the 24 h time point. Therefore, the result in Fig. 1D shown in the original manuscript has been omitted, and replaced with new data for Huh7-ACE2 cells and Vero-TMPRSS2 cells in Fig. S7D-G, in the revised manuscript. Conversely, since the effect of the Δ ORF6 virus has been replicated in the hamster models, suggesting that loss of ORF6 is more prominent in animals rather than culture cells, all data for the animal experiments have been moved to Fig. 6. Regarding the data presentation of the hamster models, measurements of viral RNA typically reflect the infection status in animal. Also, the body weight and the histochemical data support that the deletion of ORF6 influenced viral replication in animal.

5) Fig. 2E and F: GAS luciferase is needed. Fig. 2F has a big SD as indicated by error bars and the experiment should be repeated.

Answer 5

To respond to the reviewer's request, we collected new luciferase data for GAS (Fig. 1E in the revised manuscript). In addition, graphs with raw luciferase values are represented in Fig. S1A (GAS) and B (ISRE) in the revised manuscript, as requested by Reviewer #3.

6) Fig. 3E and lines 176-178: STAT1 KO or KD cells should be used.

Answer 6

Following the reviewer's suggestion, STAT1 was knocked down in Huh7 cells using two independent siRNAs, and then the luciferase assay was performed in presence of AcGFP-ORF6. Fig. 2D and E in the revised manuscript showed data for western blotting and a replication assay, respectively. The results indicate that ORF6 apparently affects the viral replication through the STAT1 signal pathway.

7) Several lines of experiments should be performed to strengthen the interaction between ORF6 and importin α 1: a) Co-immunoprecipitation is required in addition to GST pull-down in Fig. 5D. b) GFP only control is required for Fig. 5E, 5F and S3C. c) Nuclear import of importin α 1 was unaffected by ORF6 (Fig. 5E and F). Whether ORF6 affects RanGTP/CAS/importin α 1 export should be tested (ORF6 reduces cytoplasmic CAS, an importin α 1 exporter (Fig. S3E)). d) The effect of ORF6 on endogenous importin α 1 is weak at best (Fig. S3A and S3B). It is required to evaluate importin α 1 distribution during IFN- γ treatment or infection e.g. with Δ ORF6 virus. e) Is it STAT1 dependent? (ORF6-M2 and -M3 can be used as suggested in point 2 above).

Answer 7

Following the reviewer's requests, we provided the additional data as follows;

- a) As described in comment #1, we performed co-immunoprecipitation to show the binding of ORF6 to importins α 1 or α 5 (Fig. 4D in the revised manuscript). We found that ORF6 has different binding modes for each importin α subtype.
- b) AcGFP control data are presented in Fig. 4E and F, in the revised manuscript. Instead of Fig. S3C in the original manuscript, which shows subcellular distribution of endogenous importin α 1 in ORF6-transfected cells treated with H₂O₂, new results of both AcGFP and AcGFP-ORF6 are presented in Fig. S5A and B, in the revised manuscript.
- c) In Fig. S5C and D in the revised manuscript, we present new evidence that Flag-STAT1 is still localized in the cytoplasm under the H₂O₂ treatment, which indicates that importin α 1 altered the distribution from the cytoplasm to the nucleus (Flag-importin α 1 [Fig. 4E and F] and endogenous importin α 1 [Fig. S5A and B], in the revised manuscript). The data indicate that importin α 1 can shuttle between the nucleus and the cytoplasm in ORF6-transfected cells, suggesting nuclear import of importin α 1 is not affected by ORF6.

Regarding the subcellular distribution of CAS, we calculated the fluorescence intensity of CAS and provided a new graph in Fig. S6C and D in the revised manuscript. Interestingly, the nuclear population of CAS was significantly increased in ORF6 expressed cells, while importin β 1 was not dramatically shifted into the nucleus. This suggests that ORF6 could involve in the intracellular dynamics of CAS itself and/or the importin α /CAS/RanGTP complex as the reviewer hypothesized. Several reports have suggested that ORF6 binds to nucleoporins Nup98 and RAE1 to affect the nucleocytoplasmic transport, particular in the nuclear export pathway^{2,3,4}. Therefore, the nuclear export of CAS may be influenced by the interaction of ORF6 with those nucleoporins. However, such reports do not address the question of why ORF6 specifically inhibits the STAT1 signal pathway rather than all importin α/β pathways. This is a next part for us to address, whether increased level of the nuclear CAS enhances the cytoplasmic retention of importin α s in the ORF6 expressed cells. We appreciate the reviewer raising this interesting suggestion.

- d) We understand the reviewer's concern because endogenous importin α 1 already seems to be localized in the cytoplasm in the absence of ORF6, while it is shuttling between the nucleus and the cytoplasm. Therefore, to show the ORF6's effect more clearly, we used exogenous Flag-tagged importin α 1, which shows remarkable nuclear localization.
- e) We have added new evidence that no statistical changes were observed for the subcellular localization of endogenous importin α 1 by the stimulation of IFN- γ and IFN- β (Fig. S4C, D in the revised manuscript), as well as in the presence of ORF6 (Fig. S4E). In addition, our IP result (presented in Fig. 4D in the revised manuscript) supports the previous finding that STAT1 is recognized by importin α 5 (referred to as NPI-1 in the original paper), not importin α 1 (referred to as Rch1)¹. Thus, the global subcellular distribution of importin α 1 seems not to be dramatically affected by the interferon stimulation, meaning that the significant cytoplasmic distribution of importin α 1 is caused by ORF6, not STAT1. We thank the reviewer for the question about the importin α distribution under IFN stimulation.

8) Importin α 1 knockout only had minimal effect on SARS-CoV-2 replication (Fig. 6J and K): a) Can a more pronounced effect be observed with Δ ORF6 virus? b) Importin α 3, α 4, α 6, and α 8 may have substituted the role of importin α 1 (Fig. S2). Chemical inhibitors (e.g. Ivermectin) might be used.

Answer 8

Thank you for the critical suggestion. As described in the answer No.4, in this revised experiment, our Δ ORF6 virus did not show the replication effects which decrease the viral replication in culture cells (Fig. S7D-G in this revised manuscript). Hence, we discontinued usage of the recombinant virus as it was impossible to respond to the reviewer's suggestion. We apologize for the unrepresented data.

As per the reviewers' comment, the KO effect of the KPNA2 gene was limited for viral replication. To verify that the loss of KPNA2 increases the viral replication, we established KO cells again and examined the viral RNA levels and viral titer at 24 h following infection. The result indicated

that the viral titer was only statistically significant, while both viral RNA levels and viral titer tended to be increased 24 h following infection (Reviewer Only Fig. RO4B and C). In addition, we attempted to use the replicon system that transfected into the KPNA2-KO cells with AcGFP-ORF6. However, the result revealed no significant change in viral replication in the KO cells, even in the presence of ORF6 (Reviewer Only Fig. RO4D). These data suggest that the effect of ORF6 on importin α 1 function is not considered to be significant in the viral replication. We cannot exclude the possibility that importin α 1 involves in the suppression of the viral life cycle through the nuclear trafficking function. However, our data provide less critical evidence for the functional significance of importin α 1 in this viral infection disease. Therefore, we judged to omit the KPNA2-KO result parts including the expression profiling from this revised manuscript. Further studies need to be conducted to address what archives the suppressive function of importin α 1 for the viral replication.

Usage of ivermectin suggested by the reviewer showed no effect on the STAT1 distribution (Reviewer Only Fig. RO3 in the revised manuscript), even if the drug has highly specific for importin α 1 than the other subtypes. This result may support the reviewer's next suggestion about the functional compensation of importin α subtypes. In fact, the closely related subtype importin α 8 can transport cargo into the nucleus with an efficiency similar for importin α 1⁵, and remarkable nuclear exclusion was observed in ORF6-expressed cells (Fig. S3 in the revised manuscript). However, the importin α s are expressed in different tissues in human and rodent, for example importin α 1, is in highly expressed in testis or spleen, in contrast to importin α 8 expression in ovary or embryo. Such expression patterns of importin α subtypes may be critical for the functional expression of ORF6, and may be related to the observations in the hamster models. A more in-depth understanding is required for how the SARS-CoV-2 infection collapses the functionality of importin α s, with or without ORF6. Thank you for the interesting suggestion.

9) *Subcellular fractionation experiment is required to support some critical staining data e.g. for STAT1, importin α 1 and α 5.*

Answer 8

Although we have spent a lot of time attempting to address this request, we could not complete the experiments based on the molecular characteristics of the ORF6 protein. A recent paper has reported that ORF6 exhibits high toxicity in cells, which showed nearly 50% of loss in HEK293T cells upon transient transfection⁶. Indeed, we observed the AcGFP-ORF6 expressed cells at approximately 10% in whole cells in contrast to 50–60% for AcGFP itself, using a transient transfection reagent, such as Lipofectamine. To increase the expression levels of AcGFP-ORF6 and align it to the level of AcGFP, we tried to use an electroporation system (Amaxa Nucleofector) with 10 to 20-times higher amounts of AcGFP-ORF6 plasmid than the AcGFP plasmid. However, our fractionation assay could not reflect the reduction of PY-STAT in the nucleus of the ORF6-expressed cells (Reviewer Only Fig. RO2A in the revised manuscript). Results similar to STAT1 were observed for p65/RelA and HIF-1 α (Fig.

RO2B and C). Therefore, we conclude that the immunofluorescence observation most likely reflects the proper phenomena of ORF6 rather than the biochemical fractionation for bulk lysates.

Also, biochemical fractionation does not reflect the distribution of importin α s, based on our experience. For example, although endogenous importin α s, such as importin α 1, is mainly observed in the cytoplasm (Fig. S4 in the revised manuscript), the biochemical fractionation showed high presence in the nuclear fraction because importin α is shuttling between the nucleus and the cytoplasm. In addition, importin α s migrates into the nucleus upon several physiological stresses⁷, suggesting that the cell collection method may induce any stresses. Consequently, we propose that immunofluorescence observation is the optimal approach of exploring the accurate distribution of ORF6 and importin α s, rather than the biochemical fractionation assay. Thank you for the suggestion.

Reviewer #2 (Remarks to the Author):

Comments/ concerns:

1. In figure 1D, the growth differences between the wt virus and Δ ORF6 should be shown for several times post infection. Viral load on additional days post infection should also be added to figure 1F. Are there significant differences in virus induced lung pathology in hamsters infected with Δ ORF6 or WT virus?

Answer 1

In response to a suggestion by the Reviewer#2 as well as Reviewer#1, we performed infection assay again with several time points, using WT and Δ ORF6 virus. However, the results did not replicate the reduction of viral replication, in contrast to in the original data. We had repeated the experiment at several time points using not only Huh7-ACE2 cells but also VeroE6-TMPRSS2 cells. Finally, we conclude that loss of ORF6 has less influence on viral replication in cultured cells. Conversely, the Δ ORF6 virus apparently reduced in the hamster models, suggesting that loss of ORF6 could be more effective in animal than in culture cells. Regarding the lung pathology in hamsters, the data demonstrated that cells infected with Δ ORF6, which were recognized by detection of the SARS-CoV-2 NP protein, were clearly reduced in lungs in contrast to in WT, indicating that loss of ORF6 remarkably influences viral replication of the animal models.

2. In the Luc-2a-ORF6 virus, is ORF6 expression the same as in true wt virus?

Answer 2

To confirm the expression level of ORF6 in the recombinant virus, we performed western blotting for the Luc-2a-ORF6 virus along with the wild virus from NIID. Fig. S7C demonstrates that the levels of ORF6 in the recombinant virus are slightly lower but apparently detectable compared to the wild virus.

3. In fig 4E, could importin α 1 be part of complex with STAT1/ ORF6? Is it possible that there is minimal stimulation of the IFN response via transfection? Is the STAT1 pulled down clearly not phosphorylated? (This is suggested in the discussion but not shown). Does the reciprocal pull down

(IP: HA) also show this interaction?

Answer 3

Thank you for the critical suggestion with regard to the binding status of ORF6, STAT1, and importin α s. In response to the reviewer's concern, we have provided new co-immunoprecipitation data in the revised manuscript. First, as shown in Fig. 3E in the revised manuscript, AcGFP-ORF6 was co-precipitated by Flag-STAT1 using anti-Flag antibody with or without the IFN stimulation. This supports our proposal that ORF6 directly interacts with STAT1 even in the absence of IFN stimulation.

Another co-IP indicated that AcGFP-ORF6 was pulled down with Flag-importin α 1 in the unstimulated situation (showed as Untreated, Fig. 4D in the revised manuscript). This suggests that ORF6 does not compete with importin α 1 for binding with STAT1, and ORF6 separately inhibits the nuclear transport of STAT1 in a manner independent of importin α 1.

In contrast, the binding status of ORF6 and importin α 5 highly depended on the IFN stimulation. In the unstimulated condition, AcGFP-ORF6 was certainly but faintly detected in the sample precipitated by Flag-importin α 5. This corresponds to the partial distribution change of importin α 5 observed in Fig. 4A and B in the revised manuscript. Conversely, with the IFN stimulation, the band intensity of AcGFP-ORF6 was increased (the right lane in Fig. 4D). The result corresponds to the previous report that importin α 5 (referred to as NPI-1 in the original paper), not importin α 1 (referred to as Rch1) recognizes PY-STAT1 for transport into the nucleus¹. Although the evidence suggests that ORF6 is a part of a complex with importin α 5/PY-STAT1, we have no clue on how the nuclear trafficking of the complex inhibits by the binding with ORF6 at this moment.

Overall, this is a first piece of evidence that enhances our understanding how ORF6 specifically disrupts the STAT1-signaling pathway in contrast to the importin α 1-mediated nuclear trafficking. The reviewer's suggestion strengthens this paper with regard to understanding the binding status of ORF6 to STAT1 and importin α s, in the presence or absence of IFN stimulation. Thank you very much again for the critical comments.

4. Fig. 4 F and G: Does the ORF6 with the C-terminal deletion fail to interact with STAT1?

Answer 4

Following the reviewer's suggestion, we newly constructed a C-terminus deletion mutant of ORF6, and then examined the binding with STAT1. In the data presented in Fig. 3H in the revised manuscript, the interaction of the ORF6 Δ C mutant with STAT1 was apparently reduced, indicating that ORF6 directly binds to STAT1 through the C-terminal sequence.

5. Fig. 6: Knockout of importin α 1/ KPNA2 should be verified by western blot.

Answer 5

Western blot data for KPNA2 knockout is provided in Reviewer Only Fig. RO4A in the revised manuscript

6. The effect of the KPNA2 knockout on virus replication is quite modest, at best. Does the KPNA2 knockout rescue the Δ ORF6 phenotype? This might be expected to show a greater effect? What is the effect of a STAT1 knockout on wt and Δ ORF6 viruses?

Answer 6

Thank you for the critical suggestion. As mentioned above in comment No.4 by Reviewer#1, our Δ ORF6 virus failed to decrease the viral replication in culture cells in the revised manuscript (Fig. S7D-G in the revised manuscript). Therefore, we discontinued use of the recombinant virus. We apologize for the unrepresented data.

As described above in comment NO 8 by Reviewer#1, we established KO cells again and examined the viral RNA levels and viral titer at 24 h following infection. However, the viral titer was only statistically significant, while both viral RNA levels and viral titer tended to be increased 24 h following infection (Reviewer Only Fig. RO4B and C). In addition, the replicon transfected into the KPNA2-KO cells with AcGFP-ORF6 revealed no significant change in viral replication in the KO cells, even in the presence of ORF6 (Reviewer Only Fig. RO4D). These data indicate that the effect of ORF6 on importin α 1 function is not considered to be significant in the viral replication. We cannot exclude the possibility that importin α 1 may involve in the suppression of the viral life cycle through the nuclear trafficking function. However, our data provide less critical evidence for the functional significance of importin α 1 in this viral infection disease. Therefore, we judged to omit the KPNA2-KO result parts including the expression profiling from this revised manuscript. Further studies need to be conducted to address what archives the suppressive function of importin α 1 for the viral replication.

7. Fig 6h does not appear to support an effect on p65 nuclear localization – could this be shown another way? (e.g. ORF6 effect on NF- κ B responsive reporter, western blotting of nuclear vs. cytoplasmic fractions)?

Answer 7

We agree the reviewer's concern that representation of the p65 nuclear localization inhibition by ORF6 is modest compared to that of HIF-1 α in the original manuscript. Therefore, the p65 images were replaced with more clear ones (Fig. 5F in the revised manuscript).

Regarding the nuclear-cytoplasmic fractionation assay, as described to Reviewer#1 No.9 comment, we could not collect data reflecting the immunofluorescence images because of the protein character of ORF6 itself, in particular the cytotoxicity that induces cell death⁶, thereby it was difficult to align the expression levels of ORF6 to that of the control AcGFP. Although we have used an electroporation system (Amaxa Nucleofector) with 10 to 20-times higher amounts of the AcGFP-ORF6 plasmid than the AcGFP plasmid, the nuclear fraction appeared to have abundant proteins even in PY-STAT1 (Reviewer Only Fig. RO2A in the revised manuscript). Results similar to that of STAT1

were also observed for p65/RelA and HIF-1 α . Overall, we conclude that immunofluorescence observation is the best approach of showing the effect of ORF6 for the target proteins.

8. Does SARS-CoV-2 infection affect NF- κ B or HIF-1 α signaling?

Answer 8

Following the reviewer's suggestion, we infected Vero-TMPRSS2 cells with SARS-CoV-2 to observe the subcellular localization of p65 in the TNF- α stimulated cells. The infected cells were detected by the expression of ORF6. As shown in Fig. 5H and I, there were no differences in the nuclear intensities of p65 with infection. Since several papers have reported that viral components such as Nsp5 or ORF7a enhances cytokine expression through activating the NF- κ B signaling pathway^{8,9}, the inhibitory effect of ORF6 may be cancelled by the other component's functions. Our data clearly define that ORF6 potentially suppresses the importin α / β 1-mediated cargo transport, such as p65 or HIF-1 α , but the inhibitory effect is more prominent for the STAT1-signaling pathway.

Minor points:

9. What is the methodology for the results in Fig. 2C and similar figures? Are only GFP+ cells counted?

Answer 9

We calculated fluorescence intensities of only AcGFP-positive cells. To clarify the methodology, the following sentences were added in the "Indirect immunofluorescence" section in the Materials and Methods of the revised manuscript; "*Using Leica Application Suite X, cells only expressing AcGFP were extracted and then the fluorescence intensities were identified at a region of interest in the nucleus, as well in the whole cells. The relative fluorescence intensity values in the nuclei against the whole cells were calculated.*" Thank you for pointing it out.

10. Fig 2E: The ISRE responds directly to STAT1/2 heterodimers induced by type I interferon. An IFN γ responsive element (GAS) should be used in this experiment.

Answer 10

We added new data for the GAS element in Fig. 1E and G in the revised manuscript. Graphs based on the raw values are also showed in Fig. S1, as requested by Reviwer#3.

11. ORF6 appears to have an effect on importin α 5, although to a lesser degree (Fig 5C), but text seems to imply there was no effect.

Answer 11

Thank you for pointing that out. A new co-immunoprecipitation assay showed that ORF6 was certainly but modestly bound to importin α 5 in the absence of the IFN stimulation (Fig. 4D in the revised manuscript). Also, the recombinant binding assay showed a faint band of importin α 5 binding with ORF6, which was actually difficult to recognize (Fig. 4C). The data indicate that the binding affinity

between ORF6 and importin $\alpha 5$ is very low, corresponding to the immunofluorescence data that the distribution change of importin $\alpha 5$ by ORF6 was partial in comparison to that of importin $\alpha 1$ (Fig. 4A and B). We have modified the relevant sentence in the results section as follows; “*The pull-down assay results indicated that ORF6 certainly bound to Flag-importin $\alpha 1$, while only a faint band was detected for Flag-importin $\alpha 5$ (Fig. 4C).*” Additionally, the final sentences are as follows; “*Overall, there are importin α subtype specificities, with ORF6 mainly binding to importin $\alpha 1$, in contrast to the low affinity for importin $\alpha 5$, corresponding to the subcellular distribution change for each importin α .*”

Reviewer #3 (Remarks to the Author):

Specific comments:

Fig.3E:

The methodology for this section is incomplete – it is not clear how the pBAC-derived replicon was delivered into the cells and how the luciferase signal was obtained. Please include this in the manuscript.

Answer 1

Thank you for the pointing that out with regard to the replicon experiment. Following the reviewer’s request, we describe it in detail in the “Transient replicon assay” section of the Materials and Methods in the revised manuscript.

Fig.2E, 2F, 3E: It would be easier to appreciate the actual differences by using raw luciferase signals (instead of normalized values). Please consider re-plotting the graphs using raw luciferase signals. Also, it is not clear how the authors computed the “relative luciferase values.” Please clarify in the text to make it easier for the audience to appreciate the findings fully.

Answer 2

In response to the reviewer’s suggestion and the editor’s recommendation, graphs with raw luciferase values are presented in Fig. S1A (GAS) and S1B (ISRE) in the revised manuscript. Regarding the description of relative luciferase values, we add information in the “Luciferase assay” section of the Materials and Methods in the revised manuscript as follows: *Relative luciferase values were calculated based on the firefly/Renilla luciferase values of AcGFP.* Thank you for the critical suggestion.

Fig.5A & 5C and Fig. S3A & 3B: please explain why endogenous importin alpha 1 seems to predominantly localize in the cytoplasm but not the FLAG-tagged version of the protein that was ectopically expressed in cells. The FLAG-tagged version is pretty much present in the nucleus. If this discrepancy were valid, it would be fair to question whether FLAG-tagging could alter the subcellular localization of these importin proteins (i.e., Fig. 5A, 5B, and Fig. S2)

Answer 3

This is an important view for distinct subcellular distribution of endogenous and exogenous importin α s, which has been observed in the past in several papers, including ours¹⁰. Our understanding is that the differences arise from the expression levels of importin α s. Under high expression, such as in transfection or cancer tissues, its localization is mainly in the nucleus, in contrast to cytoplasmic localization in a case of low expression. As a reference, we provide images of overexpressed EGFP fused importin α 1 and α 5, which were apparently observed in the nucleus, while some lower levels of expressed proteins were observed more in the cytoplasm (Reviewer Only Fig. RO1). This indicates that the distinct distribution of importin α s depends on the expression levels, and not on fusion tag. Although the actual mechanism via which expression levels are linked to importin α s distribution remain unclear, nuclear-cytoplasmic recycling balance of importin α may be impaired by high expression^{7,11}. In conclusion, the subcellular distribution of importin α s depends on expression levels, not the fusion tags.

Fig. 6J and 6K: please provide evidence to show that KPNA2 is being CRISPR knocked out (e.g., WB or IFA; should be pretty straightforward as the authors have presented immunofluorescent imaging for KPNA2 in Fig. S3). Please include results for validating the knockout(s). How do you know that these are “on-target” knockouts? It is common to include a “cDNA addback” to see if the phenotype could be rescued. Please include the rescue experiment (should be straightforward as the KPNA2 cDNA was used in Fig. 5). Also, it is not clear if one sgRNA or three sgRNAs were used to edit KPNA2 (see line 276 vs. line 509). Please clarify.

Answer 4

Following the reviewer’s suggestion, we have provided western blot data on KPNA2 knockout in Reviewer Only Fig. RO4A, in the revised manuscript. Regarding the rescue experiment, it has been difficult to addback KPNA2 cDNA, since the majority of KO cells died after infection.

As described above in comment No.4 by Reviewer#1, the newly established KPNA2 KO cells showed the modest effect on the viral replication again, and the viral titer was only statistically significant (Reviewer Only Fig. RO4B and C). In addition, the replicon assay demonstrated no significant reduction in viral replication in the KO cells, even in the presence of ORF6 (Reviewer Only Fig. RO4D). These data suggests that the effect of ORF6 on importin α 1 function is not considered to be significant in the viral replication. Although we are interested in the involvement of importin α 1 to suppress the viral life cycle through the nuclear trafficking function, we judged to remove the KPNA2 KO result parts including the expression profiling from this revised manuscript.

References

1. Sekimoto T, Imamoto N, Nakajima K, Hirano T, Yoneda Y. Extracellular signal-dependent nuclear import of Stat1 is mediated by nuclear pore-targeting complex formation with NPI-1, but not Rch1. *EMBO J* **16**, 7067-7077 (1997).

2. Miorin L, *et al.* SARS-CoV-2 Orf6 hijacks Nup98 to block STAT nuclear import and antagonize interferon signaling. *Proc Natl Acad Sci USA* **117**, 28344-28354 (2020).
3. Addetia A, *et al.* SARS-CoV-2 ORF6 Disrupts Bidirectional Nucleocytoplasmic Transport through Interactions with Rae1 and Nup98. *mBio* **12**, (2021).
4. Kato K, *et al.* Overexpression of SARS-CoV-2 protein ORF6 dislocates RAE1 and NUP98 from the nuclear pore complex. *Biochem Biophys Res Commun* **536**, 59-66 (2021).
5. Kimoto C, *et al.* Functional characterization of importin α 8 as a classical nuclear localization signal receptor. *Biochim Biophys Acta* **1853**, 2676-2683 (2015).
6. Lee JG, Huang W, Lee H, van de Leemput J, Kane MA, Han Z. Characterization of SARS-CoV-2 proteins reveals Orf6 pathogenicity, subcellular localization, host interactions and attenuation by Selinexor. *Cell Biosci* **11**, 58 (2021).
7. Miyamoto Y, *et al.* Cellular stresses induce the nuclear accumulation of importin α and cause a conventional nuclear import block. *J Cell Biol* **165**, 617-623 (2004).
8. Li W, *et al.* SARS-CoV-2 Nsp5 Activates NF- κ B Pathway by Upregulating SUMOylation of MAVS. *Frontiers in immunology* **12**, 750969 (2021).
9. Su CM, Wang L, Yoo D. Activation of NF- κ B and induction of proinflammatory cytokine expressions mediated by ORF7a protein of SARS-CoV-2. *Scientific reports* **11**, 13464 (2021).
10. Yasuda Y, *et al.* Nuclear retention of importin α coordinates cell fate through changes in gene expression. *EMBO J* **31**, 83-94 (2012).
11. Miyamoto Y, *et al.* Importin α can migrate into the nucleus in an importin β - and Ran-independent manner. *EMBO J* **21**, 5833-5842 (2002).

Reviewers' comments:

Reviewer #1 (Remarks to the Author):

I am satisfied with the revisions made.

Reviewer #2 (Remarks to the Author):

This is a revised version of a previously submitted manuscript focusing on the role of SARS-CoV2 ORF6. As noted in the introduction, the role of ORF6 in inhibition of STAT1-dependent interferon signaling has been extensively reported on. In this study, the authors demonstrate direct binding of STAT1 to ORF6, which results in STAT1 retention in the cytoplasm in the presence of type I or II interferons. The direct interaction of ORF6 and STAT1 is a novel aspect of this study, as other studies have emphasized the interaction of ORF6 with importins and/ or nuclear pore components in blocking nuclear translocation of STAT1. The manuscript would be strengthened by additional experiments (discussed below) demonstrating that mutants without STAT1 without the ability to sequester STAT1 (M2 and M3) also have reduced STAT1 binding activity.

Comments/ concerns:

1. The data showing interaction between ORF6 and STAT1 relies heavily on overexpression of tagged proteins. Does ORF6 pulldown endogenous STAT1?
2. The M2 and M3 mutants show an inability to prevent nuclear localization of STAT1 in response to IFN γ (Fig. S2). These data are important enough to the paper's conclusions that it would be preferable to include them in the main text, rather than as supplemental data. Do these mutants also fail to pull down STAT1 in the immunoprecipitation or GST pulldown assay? (The ORF6 Δ C mutant seems to retain some binding for STAT1 (Fig. 3H), so it is not clear how much binding will be reduced for M2 and M3). This information would be of more value in figure 3 than the ORF6 Δ 9 data, which seems irrelevant to this study.
3. Does the C-terminal domain of ORF6 (i.e. the domain in the M0 mutant) affect interferon signaling/ STAT1 nuclear localization? (If used as a GFP fusion, one might expect that if this domain mediates binding, but not retention, the GFP signal may be shifted to the nucleus).
4. The text (lines 196-197) suggests that ORF6 has little effect on Importin- α 5, but Fig 4B indicates a significant ($P < 0.001$) change in Imp- α 5 in the presence of ORF6. Although less than observed with Imp- α 1, it seems that this result should not be dismissed as "mainly retained in the nucleus." The possible presence of STAT1 in the Imp- α 5/ORF6 complex (4D) also indicates that this interaction may play a role in STAT1 cytoplasmic sequestration.
5. Is the replicon described in figure 2 differentially affected by interferon (added to culture) in cells co-transfected with GFP-ORF6, as compared to GFP alone? The authors also note that, when re-examining an ORF6 deletion virus, they did not see significantly different growth kinetics compared to wild-type. Is the Δ ORF6 virus more sensitive to exogenous interferon?
6. Line 334-335: "we could not determine how STAT1 and importin α 1 bind simultaneously to the C-terminus of ORF6."

Minor points:

7. Lines 315-316 indicate state "ORF6 negatively regulates the nuclear import of HIF α and NF- κ B p65..." although figure 5I shows no significant difference in p65 nuclear localization.

Response to Reviewer #2 (Remarks to the Author):

Comments/ concerns:

1. The data showing interaction between ORF6 and STAT1 relies heavily on overexpression of tagged proteins. Does ORF6 pulldown endogenous STAT1?

Answer 1

According to the reviewer's concern, we performed a pull-down assay to show the binding of endogenous STAT1 with ORF6. HeLa cells were treated with IFN- γ , and then, cell lysates were incubated with GST-GFP-ORF6 immobilized to GST-beads. GST-GFP was used as a negative control. Fig. 4D in the re-revised manuscript reveals that endogenous PY-STAT1 clearly bound to GST-GFP-ORF6 but not GST-GFP.

2. The M2 and M3 mutants show an inability to prevent nuclear localization of STAT1 in response to IFN γ (Fig. S2). These data are important enough to the paper's conclusions that it would be preferable to include them in the main text, rather than as supplemental data. Do these mutants also fail to pull down STAT1 in the immunoprecipitation or GST pulldown assay? (The ORF6 Δ C mutant seems to retain some binding for STAT1 (Fig. 3H), so it is not clear how much binding will be reduced for M2 and M3). This information would be of more value in figure 3 than the ORF6 Δ 9 data, which seems irrelevant to this study.

Answer 2

As per the reviewer's suggestion, Fig. S2A and B in the prior revised manuscript was moved to Fig. 2B and C in the re-revised manuscript.

To further analyze the binding of the C-terminus of ORF6 to STAT1, we produced GST-M1-GFP, GST-M2-GFP, and GST-M3-GFP recombinant proteins, as well as GST-M0-GFP, and then performed GST-pull down assays for endogenous PY-STAT1. As shown in Fig. 4E in the re-revised manuscript, PY-STAT1 apparently bound to the M0 and M1 proteins, consistent with the inhibitory effect on the nuclear translocation of STAT1 (Fig. 2B in the re-revised manuscript). In contrast, the M3 protein completely lost its binding properties. For the M2 protein, binding properties were almost abolished, but subtle binding was apparent, whereas it was difficult to judge whether the faint band was indicative of the specific or non-specific binding properties of M2. Regarding the faint band for the C-terminus-deleted mutant of ORF6 (Fig. 4C in the re-revised manuscript), we assume that although the deletion mutant had lost its binding properties for STAT1, conformational or functional abnormality of the protein based on deletion of the C-terminus could have resulted in non-specific binding to STAT1.

Considering the limitation of the biochemical analysis, we concluded that deletion of the C-terminus abrogated the binding to STAT1.

3. Does the C-terminal domain of ORF6 (i.e. the domain in the M0 mutant) affect interferon signaling/ STAT1 nuclear localization? (If used as a GFP fusion, one might expect that if this domain mediates binding, but not retention, the GFP signal may be shifted to the nucleus).

Answer 3

In response to the reviewer's suggestion, we observed the subcellular distribution of the ORF6 C-terminal sequence (GST-GFP fused M0–M3 mutants) when the proteins were microinjected into the cytoplasm of Huh7 cells, followed by stimulation with IFN- γ . Under unstimulated conditions, GST-M0-GFP, GST-M1-GFP, and GST-M2-GFP localized not only to the cytoplasm but also to the nucleus (Fig. 4F in the re-revised manuscript). In contrast, the GST-M3-GFP protein was only present in the cytoplasm, as with the GST-GFP control protein. Upon IFN stimulation, the subcellular localization of all proteins was not affected, as in unstimulated conditions, and the inhibition of PY-STAT1 nuclear transport was observed in the M0- and M1-injected cells (Fig. 4G). These results yield two important findings. First, only the C-terminal sequence of ORF6 has the ability to inhibit the nuclear localization of STAT1, whereas it could not completely achieve this nuclear exclusion, unlike the full-length ORF6 protein. Second, the C-terminal sequence, and in particular residues 56–61, which is the M3 region, has a nuclear localization signal (NLS), because the GST-GFP protein is known to be a carrier protein that cannot migrate to the nucleus and the M3 mutant lost its nuclear migration.

Notably, the M2 mutant lost its inhibitory effect on STAT1 nuclear localization. Specifically, this reveals that if the C-terminus functions as an NLS recognized by importin α and followed by transportation into the nucleus, the binding with importin α could not directly be associated with the nuclear exclusion of STAT1. This supports our proposal that ORF6 separately influences the STAT1 signaling pathway and the importin α -mediated pathway. Recently, it has been reported that a methionine at the 58 position in ORF6, which is contained in the M3 region, is critical for binding to Nup98¹; however, this is inconsistent as the M2 mutant lost its inhibitory effect on STAT1 nuclear localization. In this re-revised manuscript, we demonstrated that STAT1 lost its binding not only to the M3 protein but also to the M2 protein (Fig. 4E in the re-revised manuscript). These findings indicate that the nuclear exclusion of STAT1 is mainly achieved via direct binding with ORF6 through the C-terminus at residues 53–61, but we could not exclude the possibility that binding with importin α and Nup98 also contributes to the trafficking

inhibition of STAT1. The reviewer's suggestion has strengthened our understanding of the molecular mechanism underlying the binding of ORF6 with either importin α , Nup98, or STAT1. We thank the reviewer for providing this interesting suggestion.

4. The text (lines 196-197) suggests that ORF6 has little effect on Importin- α 5, but Fig 4B indicates a significant ($P < 0.001$) change in Imp- α 5 in the presence of ORF6. Although less than observed with Imp- α 1, it seems that this result should not be dismissed as “mainly retained in the nucleus.” The possible presence of STAT1 in the Imp- α 5/ORF6 complex (4D) also indicates that this interaction may play a role in STAT1 cytoplasmic sequestration.

Answer 4

To avoid confusion for the reader, the sentence “the Flag-importin α 5 was mainly retained in the nucleus” was changed to “*the number of cells in which Flag-importin α 5 is localized in the cytoplasm was limited compared with that of Flag-importin α 1*”.

We could not rule out the possibility that ORF6 affects the interaction between importin α 5 and STAT1. However, our data in Fig. 5 in the revised manuscript demonstrate that the binding of ORF6 to importin α 5 is not a primary factor inhibiting STAT1 nuclear localization.

5. Is the replicon described in figure 2 differentially affected by interferon (added to culture) in cells co-transfected with GFP-ORF6, as compared to GFP alone? The authors also note that, when re-examining an ORF6 deletion virus, they did not see significantly different growth kinetics compared to wild-type. Is the Δ ORF6 virus more sensitive to exogenous interferon?

Answer 5

We thank the reviewer for pointing out this important aspect to understand the physiological significance of ORF6 in viral replication. Following the reviewer's suggestion, we used Vero E6 replicon stable cell lines ² to examine the effect of ORF6 for the viral replication. As shown in Fig. S2 in the re-revised manuscript, the luciferase values of the replicon are decreased in response to the IFN- β stimulation and it was increased by the presence of ORF6 (Fig. S2 in the re-revised manuscript). Thus, our data indicate that the addition of ORF6 could rescue the replicon activity by inhibiting IFN signaling.

On the other hand, understanding the phenotype of recombinant viruses is more difficult than understanding the replicon. Several components of SARS-CoV-2, such as nsp1, ORF3b, ORF7, and M protein, have been shown to antagonize IFN signaling,

indicating that multiple factors are mutually involved in the virus lifecycle by antagonizing IFN stimulation. Further study is needed to understand how ORF6 functions in the virus lifecycle.

6. Line 334-335: “we could not determine how STAT1 and importin α 1 bind simultaneously to the C-terminus of ORF6.”

Answer 6

Our new data shown in Fig. 4F and G in the re-revised manuscript revealed that the nuclear distribution of the C-terminal mutants is not correlated with STAT1 nuclear exclusion. Since this sentence might be confusing for the reader, we have omitted it from the re-revised manuscript.

Minor points:

7. Lines 315-316 indicate state “ORF6 negatively regulates the nuclear import of HIF1 α and NF- κ B p65...,” although figure 5I shows no significant difference in p65 nuclear localization.

Answer 7

Our data revealed that ORF6 potentially exerts inhibitory effects on the nuclear import of cargo proteins such as HIF1 α and NF- κ B p65 (Fig. 6A-G in the re-revised manuscript). However, the inhibitory effect on p65 was not clearly shown in virus-infected cells (Fig. 6J and K). Since it has been shown that several viral components, such as nsp5, can enhance cytokine expression by activating the NF- κ B signaling pathway³, it could conversely function for ORF6. We have already mentioned this in the Discussion section as follows; “*Since several recent papers have reported that viral components, such as Nsp5 or ORF7a, enhances cytokine expression through activating the NF- κ B signaling pathway^{42, 43}, the inhibitory effect of ORF6 may be counteracted by the other components. Overall, at least in the classical nuclear import pathway, the inhibitory effects of ORF6 could be limited, in contrast to the specificity for the STAT1-signaling pathway.*”

References

1. Miorin L, *et al.* SARS-CoV-2 Orf6 hijacks Nup98 to block STAT nuclear import and antagonize interferon signaling. *Proc Natl Acad Sci U S A* **117**, 28344-28354 (2020).
2. Tanaka T, *et al.* Establishment of a stable SARS-CoV-2 replicon system for application in high-throughput screening. *Antiviral research* **199**, 105268 (2022).

3. Li W, *et al.* SARS-CoV-2 Nsp5 Activates NF- κ B Pathway by Upregulating SUMOylation of MAVS. *Frontiers in immunology* **12**, 750969 (2021).

REVIEWERS' COMMENTS:

Reviewer #2 (Remarks to the Author):

My concerns have been adequately addressed.